# An extrinsic motor directs chromatin loop formation by cohesin

Thomas M Guérin [ID][1,2], Christopher Barrington[3], Georgii Pobegalov [ID][4,5], Maxim I Molodtsov[4,5] & Frank Uhlmann [ID][1✉]

## Abstract

The ring-shaped cohesin complex topologically entraps two DNA molecules to establish sister chromatid cohesion. Cohesin also shapes the interphase chromatin landscape with wide-ranging implications for gene regulation, and cohesin is thought to achieve this by actively extruding DNA loops without topologically entrapping DNA. The 'loop extrusion' hypothesis finds motivation from in vitro observations—whether this process underlies in vivo chromatin loop formation remains untested. Here, using the budding yeast *S. cerevisiae*, we generate cohesin variants that have lost their ability to extrude DNA loops but retain their ability to topologically entrap DNA. Analysis of these variants suggests that in vivo chromatin loops form independently of loop extrusion. Instead, we find that transcription promotes loop formation, and acts as an extrinsic motor that expands these loops and defines their ultimate positions. Our results necessitate a re-evaluation of the loop extrusion hypothesis. We propose that cohesin, akin to sister chromatid cohesion establishment at replication forks, forms chromatin loops by DNA–DNA capture at places of transcription, thus unifying cohesin's two roles in chromosome segregation and interphase genome organisation.

**Keywords** Cohesin; Loop Capture; Loop Extrusion; Transcription; SMC Complexes
**Subject Categories** Cell Cycle; Chromatin, Transcription & Genomics; DNA Replication, Recombination & Repair

## Introduction

Cohesin was first identified for its role in sister chromatid cohesion, holding DNA replication products together to allow their faithful distribution to daughter cells during cell divisions (Guacci et al, 1997; Losada et al, 1998; Michaelis et al, 1997). The ring-shaped cohesin complex achieves this by topologically entrapping both sister chromatids (Haering et al, 2008; Murayama et al, 2018;

Richeldi et al, 2024). At a molecular level, cohesin sequentially and topologically entraps two DNAs, with a preference for double-stranded DNA (dsDNA) followed by single-stranded DNA (ssDNA). This configuration matches that at DNA replication forks where the dsDNA leading strand product lies juxtaposed to the unwound ssDNA lagging strand. Second ssDNA capture is labile but turns stable by ssDNA to dsDNA conversion during lagging strand DNA synthesis and concomitant cohesin acetylation (Minamino et al, 2023; Murayama et al, 2018).

In addition to sister chromatid cohesion, cohesin plays a key role in interphase genome organisation. Cohesin establishes chromatin loops, as well as demarcates topologically associating domains (TADs) within which chromatin interactions are enriched (Hadjur et al, 2009; Rao et al, 2017; Schwarzer et al, 2017; Wendt et al, 2008). In mammalian genomes, cohesin-dependent loops are often seen between chromosomal CCCTC-binding factor (CTCF) binding sites, where cohesin accumulates. Acutely removing cohesin from interphase nuclei causes only limited transcriptional changes (Hsieh et al, 2022; Pauli et al, 2010; Rao et al, 2017; Schwarzer et al, 2017). At the same time, changing cohesin-dependent chromatin interactions by altering CTCF binding sites profoundly impacts on gene regulation in the longer term (Calderon et al, 2022; Kane et al, 2022; Lupiáñez et al, 2015; Nakato et al, 2023; Rinzema et al, 2022).

In principle, and in analogy to sister chromatid cohesion establishment, cohesin could form interphase chromatin loops by sequential topological capture of DNA sequences along the same chromatin chain. Indeed, we suggest in our present study that chromatin loops form by such a 'loop capture' mechanism. On the other hand, over recent years, an alternative model for loop formation has gained prominence, 'loop extrusion' (Davidson and Peters, 2021; Higashi and Uhlmann, 2022; Kim et al, 2023a; Yatskevich et al, 2019). According to the loop extrusion hypothesis, cohesin generates a small DNA loop that cohesin then enlarges using intrinsic DNA motor activity. Single-molecule in vitro experiments have strikingly illustrated cohesin-mediated DNA loop extrusion (Davidson et al, 2019; Higashi et al, 2021; Kim et al, 2019). However, in vitro loop extrusion typically requires the presence of DNA dyes or adducts that alter DNA bendability. In addition, very low external forces stall loop extrusion, and it remains uncertain how cohesin would navigate the complex in vivo

[1]Chromosome Segregation Laboratory, The Francis Crick Institute, London, UK. [2]Université Paris Cité and Université Paris-Saclay, Inserm, CEA, Stabilité Génétique Cellules Souches et Radiations, Fontenay-aux-Roses, France. [3] Bioinformatics & Biostatistics Science Technology Platform, The Francis Crick Institute, London, UK. [4]Mechanobiology and Biophysics Laboratory, The Francis Crick Institute, London, UK. [5]Department of Physics and Astronomy, University College London, London, UK. ✉E-mail: frank.uhlmann@crick.ac.uk

chromatin landscape. DNA-bound obstacles have variably been portrayed as surmountable (Pradhan et al, 2021) or as barriers (Borrie et al, 2023; Dequeker et al, 2022; Roisné-Hamelin et al, 2024). Whether in vivo chromatin loops and TADs indeed form by the in vitro observed loop extrusion mechanism remains unknown.

Support for the loop extrusion hypothesis has been derived from computational modelling. Given a range of assumptions, such models can explain observed in vivo chromatin features (Banigan et al, 2023). Notable characteristics include the 'convergence rule', by which the CTCF binding site orientation dictates the directionality of chromatin interactions. This observation is easiest explained by cohesin translocation along chromatin to probe CTCF orientation. However, whether intrinsic motion or an extrinsic motor moves cohesin rings, or chromatin moves through passive cohesin slip-links (Bonato et al, 2020) cannot be differentiated by modelling. A second characteristic of loop extrusion is the exclusive introduction of *cis*-chromatin interactions, readily explaining cohesin-dependent sister chromatid individualisation in prophase (Batty et al, 2023), and chromosome compaction following cohesin release factor Wapl depletion (Tedeschi et al, 2013). On the other hand, loop capture also occurs more likely in *cis* than in *trans*, and *cis* interactions gradually come to dominate as individualisation proceeds (Cheng et al, 2015; Tang et al, 2023).

A direct experimental test of the loop extrusion hypothesis has not yet been performed. In such a test, cohesin mutations that disrupt in vitro loop extrusion will be created, and the consequences on the in vivo chromatin landscape investigated. Here, we perform this test using the budding yeast *S. cerevisiae* model. We create an allelic series of cohesin variants that have lost loop extrusion activity to various degrees. The analysis of cells harbouring these variants suggests that in vivo chromatin loops form independently of cohesin's in vitro loop extrusion ability. Rather, loops correlate with cohesin's ability to topologically entrap DNA. We provide evidence for an alternative, unified model for cohesin function, in which cohesin rings establish both sister chromatid cohesion as well as interphase chromatin loops by sequential topological entrapment of two DNAs, in the vicinity of replication forks or sites of gene transcription.

# Results

## Loop extrusion by budding yeast cohesin

We purified recombinant budding yeast cohesin, as well as its loader, following overexpression in budding yeast (Figs. 1A and EV1A; Minamino et al, 2018). When added to 48.5 kb long λ-DNA, loosely tethered to a flow cell surface and stained with SYTOX Orange, we observed efficient DNA loop extrusion (Fig. 1B). The observed loop extrusion rate ($620 \pm 440$ bp s$^{-1}$, mean $\pm$ s.d.) was only slightly lower than previously reported rates of human and fission yeast cohesin (Fig. EV1B; Davidson et al, 2019; Higashi et al, 2021; Kim et al, 2019). Consistent with previous observations, loop extrusion by budding yeast cohesin strictly depended on the presence of the cohesin loader (Fig. 1B). Thus, the ability to perform in vitro loop extrusion extends to budding yeast cohesin.

Several molecular models have been proposed to explain loop extrusion (Bauer et al, 2021; Dekker et al, 2023; Higashi et al, 2021; Marko et al, 2019; Ryu et al, 2020). While the actual mechanism

remains to be ascertained, DNA binding by cohesin's Scc3 subunit plays a central role in all proposals. Consistent with such a contribution, charge reversal mutations in human Scc3[STAG1] render cohesin unable to extrude DNA loops, as did mutations to a DNA-binding surface on the Smc1 ATPase head (Bauer et al, 2021; Shi et al, 2020). We replaced three budding yeast Scc3 lysine residues known to contact DNA with glutamate (Scc3[3E]) (Li et al, 2018), as well as made four glutamate exchanges on Smc1 (Smc1[4E]; Fig. 1A). These alterations did not noticeably change cohesin's overall affinity to DNA, as measured in a DNA electrophoretic mobility shift assay. Scc3[3E]- and Smc1[4E]-cohesin also retained their ability to entrap DNA in an ATP-dependent, salt-resistant manner, albeit with reduced efficiency, especially in the case of Smc1[4E]-cohesin (Fig. EV1C,D). Monitoring loop extrusion in real time, while DNA molecules were stretched by mild liquid flow, revealed that Scc3[3E]-cohesin had lost its ability to extrude DNA loops, while Smc1[4E]-cohesin showed a substantially reduced loop formation frequency (Fig. 1B). To assess loop extrusion in a more sensitive assay, we incubated cohesin and its loader with DNA in the absence of flow, then applied flow solely to visualise the resultant DNA loops. In this assay, Smc1[4E]-cohesin formed DNA loops with close to half the wild-type efficiency, while Scc3[3E]-cohesin remained loop extrusion-deficient (Fig. EV1E).

The above loop extrusion assays, like those previously reported (Davidson et al, 2019; Higashi et al, 2021; Kim et al, 2019), were performed under low ionic strength conditions. To investigate loop extrusion under more physiological conditions, we raised the salt concentration to 100 mM NaCl and observed efficient loop extrusion by wild-type cohesin after adjusting the magnesium to ATP ratio in our reaction buffer. Even under these physiological ionic strength conditions, Scc3[3E]-cohesin remained loop extrusion-deficient, while Smc1[4E]-cohesin showed a substantially reduced loop formation rate (Fig. EV1F).

## Life without loop extrusion?

We next generated budding yeast strains expressing Scc3[3E] or Smc1[4E] as the sole source of these respective cohesin subunits. The resultant strains displayed no noticeable growth defects nor sensitivities to genome-damaging agents (Fig. EV2A). When arrested at G2/M (by release from α-factor synchronisation into the nocodazole-containing medium) Scc3[3E] and Smc1[4E] cells displayed slight sister chromatid cohesion defects, compared to a wild-type control (Fig. EV2B), perhaps because of the compromised ability of the mutant cohesin complexes to load onto DNA (Li et al, 2018). This interpretation found support when we measured in vivo cohesin loading using calibrated ChIP-sequencing. Scc3[3E]- and Smc1[4E]-cohesin associated with chromosomes in a pattern indistinguishable from wild-type cohesin, but at reduced levels (Figs. 1C and EV2C). These observations suggest that cohesin loop extrusion is dispensable for overall cell fitness and genome stability.

Budding yeast displays a prominent cohesin-dependent looping pattern between neighbouring cohesin binding sites, which has been visualised using micro-C (Costantino et al, 2020). We observed these chromatin loops similarly in a wild-type strain, as well as in strains harbouring loop extrusion-deficient Scc3[3E]- or Smc1[4E]-cohesin (Fig. 1C). Quantification of the loop signals (corner scores, representing the interaction differential between the centre and corner of each loop region) showed a reduced loop intensity in Scc3[3E]- and

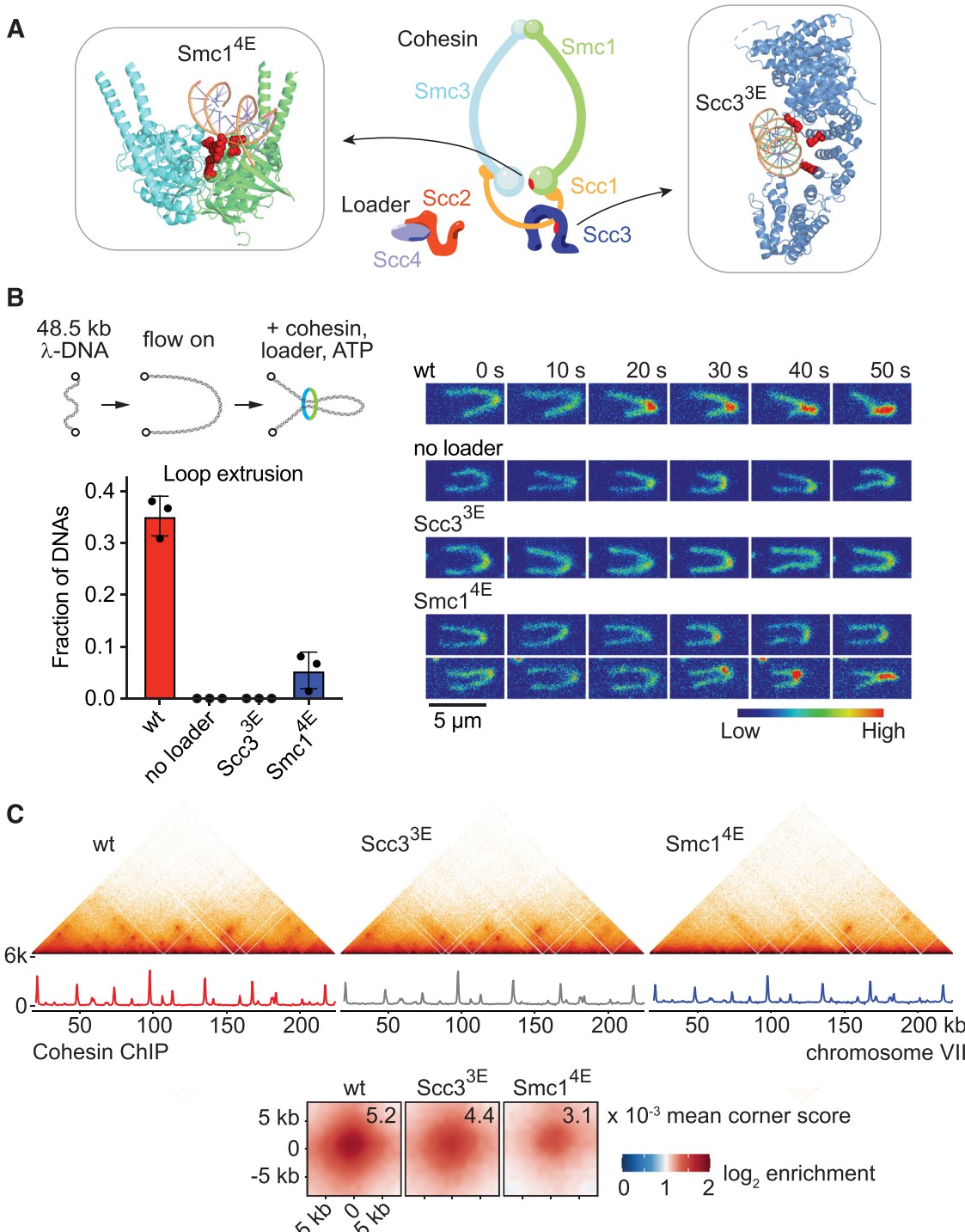

**Figure 1. Chromatin loop formation by loop extrusion-deficient cohesin.**

(A) Overview schematic of cohesin and its loader, and structures of the Scc3 (PDB: 6H8Q; Li et al, 2018) and Smc1 (PDB: 6ZZ6; Collier et al, 2020) subunits bound to DNA, highlighting the amino acids that were mutated to glutamates to generate Scc3[3E] (K423E, K520E, K669E) and Smc1[4E] (R53E, R58E, N60E, K63E). (B) Schematic representation of an in vitro loop extrusion assay. Loop extrusion efficiencies of wild-type (wt) cohesin in the presence and absence of loader, as well as of Scc3[3E]- and Smc1[4E]-cohesin, were measured in three independent repeat experiments. Individual data points are presented, bars indicate the mean and error bars the standard deviation ($n_{wt} = 201$, $n_{no\ loader} = 224$, $n_{Scc3}{}^{3E} = 301$, $n_{Smc1}{}^{4E} = 260$). Example time-lapse recordings of loop extrusion. The DNA is stained with SYTOX Orange and shown using an arbitrary linear intensity scale. Scale bar, 5 µM. (C) 500 bp-resolution merged micro-C contact maps, as well as corresponding calibrated cohesin ChIP-seq traces, from two independent experiments of G2/M arrested cells harbouring wt or loop extrusion-deficient cohesin. Cohesin ChIP used Smc3-Pk₃ in the wt and Scc3[3E] strains, or Smc1[4E]-Pk₃ with a wt Smc1-Pk₃ strain included for normalisation. Aggregate chromatin structure is shown in wt, Scc3[3E] and Smc1[4E] strains at loops detected by chromosight (Matthey-Doret et al, 2020) and linked to cohesin anchors in the wt strain ($n = 1060$). Mean corner scores are indicated. Source data are available online for this figure.

Smc1[4E]-expressing cells, as might be expected from reduced cohesin levels, but loops remained clearly discernible (Figs. 1C and EV2D). Our allelic series of cohesin variants allowed us to compare their in vitro loop extrusion and DNA capture abilities with in vivo chromatin loop formation. Scc3[3E] cohesin lacks in vitro loop extrusion while maintaining close-to-normal in vivo loops. Smc1[4E] cohesin in turn retains in vitro loop extrusion but shows weakened in vivo loops, correlating with a reduced DNA capture ability. Thus, in vivo chromatin loop formation quantitatively correlates with cohesin's DNA capture, more so than with its loop extrusion activity.

A recent report suggested that human cohesin can perform loop extrusion without its Scc3[STAG1/2] subunit (preprint: Barth et al, 2023). While we were unable to observe in vitro loop extrusion by budding yeast cohesin lacking Scc3, we nevertheless investigated whether in vivo chromatin loops could form without this subunit. We mapped chromatin loops in cells from which Scc3 could be depleted using an auxin-inducible degron. This experiment demonstrated that chromatin loop formation strictly relies on Scc3 (Fig. EV3), likely because of the subunit's importance for topological DNA entrapment (Collier et al, 2020). Taken together, the observations that in vivo chromatin loop formation depends on Scc3, yet is supported by loop extrusion-deficient Scc3[3E], suggests that chromatin loops form by a mechanism different from that observed by in vitro loop extrusion experiments.

## Meiosis without loop extrusion?

A cohesin-dependent chromatin looping pattern in budding yeast becomes especially pronounced during meiotic prophase, when cohesin forms part of the meiotic chromosome axis (Klein et al, 1999; Schalbetter et al, 2019). We therefore constructed homozygous diploid Scc3[3E] and Smc1[4E] strains and placed these under sporulation conditions to undergo meiosis. These diploids successfully completed meiotic cell divisions. Spores from the strain harbouring loop extrusion-deficient Scc3[3E] cohesin showed >90% viability, comparable to spores from a wild-type diploid, while spore viability from the SMC1[4E] strain was somewhat compromised (Fig. EV2E). This result suggests that all essential aspects of meiotic chromosome segregation, which includes meiotic axis formation, can proceed without cohesin's ability to perform DNA loop extrusion.

## In vivo loop expansion without the cohesin loader

The above experiment used cohesin variants that, while being loop extrusion-deficient under a range of in vitro conditions, retained their ability to form chromatin loops in vivo. At face value, these observations suggest that chromatin loops form by a mechanism different from loop extrusion. A caveat to this conclusion is that in vivo loop extrusion might be so much more robust that even our defective cohesin variants perform this reaction inside cells. To address this possibility, we sought an additional way to probe the contribution of loop extrusion to generating the in vivo chromatin loop pattern. We turned to the cohesin loader, an integral component of the loop extrusion mechanism (Davidson et al, 2019; Higashi et al, 2021; Kim et al, 2019).

As the cohesin loader is essential for cohesin association with chromosomes, we devised an experimental strategy in which cohesin was first loaded onto chromosomes, next the loader was removed, and only then were changes to the chromatin landscape

induced. We utilised a temperature-sensitive mutation, rat1-1, in the 5′–3′ RNA exonuclease that promotes transcription termination at polyadenylation signals (Han et al, 2023). Transcriptional readthrough after rat1-1 inactivation displaces a subset of cohesin peaks and leads to cohesin accumulation at adjacent peaks (Figs. 2A and EV4A; Ocampo-Hafalla et al, 2016). As a consequence, the micro-C pattern of rat1-1 cells at a restrictive temperature revealed many chromatin loops that are longer than those in a wild-type control (Fig. 2A,B).

The above trial compared steady-state loop sizes between wild-type and rat1-1 cells. We next investigated whether pre-existing chromatin loops change their size following rat1-1 inactivation. We arrested rat1-1 cells at a permissive temperature in G2/M, when existing chromatin loops have stabilised (Bastié et al, 2022). rat1-1 inactivation by temperature shift led to loop expansion, yielding a final loop pattern similar to that seen above (Figs. 2C and EV4B). This outcome suggests that transcriptional readthrough enlarges pre-existing chromatin loops.

We now repeated the loop expansion experiment but depleted the Scc2 cohesin loader subunit, by promoter shut-off and an auxin-inducible degron (Muñoz et al, 2019), before rat1-1 inactivation. When we similarly depleted Scc2 before cell cycle entry, no observable loops formed, confirming effective depletion (Fig. EV4C). In contrast, when we depleted Scc2 after cohesin-dependent loops had formed, rat1-1 inactivation again resulted in widespread loop expansion (Figs. 2D and EV4D). These observations suggest that cohesin loader-dependent loop extrusion is dispensable for in vivo chromatin loop growth, and that transcription acts as an extrinsic motor that expands pre-existing chromatin loops. Transcription is known to push cohesin rings along transcription units (Davidson et al, 2016; Lengronne et al, 2004; Ocampo-Hafalla et al, 2016) and loop expansion likely occurred as transcription pushed cohesins that were engaged in chromatin looping.

## Transcription promotes chromatin loop formation

If not by loop extrusion, how does cohesin initiate chromatin loop formation? Transcription has previously been implicated in establishing cohesin-dependent chromatin architectural patterns. It was proposed that these patterns arise because RNA polymerases form barriers to loop extrusion (Banigan et al, 2023; Jeppsson et al, 2022; Wike et al, 2021; Zhang et al, 2021). To revisit this phenomenon, we created a yeast strain in which both the Rpb1 and Rpb3 subunits of RNA polymerase II were fused to an FRB fragment that can be depleted from the nucleus by rapamycin addition (Fig. EV5A,B; Haruki et al, 2008). We arrested cells in late G1 phase by Cdk inhibitor Sic1 overexpression. Unlike in pheromone α-factor arrested cells when cohesin is absent, cohesin is present in Sic1-arrested cells and dynamically turns over on chromosomes due to the action of the cohesin release factor Wapl (Fig. EV5C; Lopez-Serra et al, 2013). Following transcription inhibition, loop signals between cohesin binding sites vanished (Fig. 3A), confirming a key role of transcription in establishing and maintaining the loop pattern. When we repeated the experiment in G2/M arrested cells, when cohesin acetylation has slowed cohesin turnover, loop signals weakened but remained detectable following transcription inhibition.

As a control for the looping pattern, calibrated ChIP-sequencing showed that cohesin was present at most of its usual binding sites, but peaks appeared less sharp following transcription inhibition, confirming previous observations (Fig. 3A; Jeppsson et al, 2022).

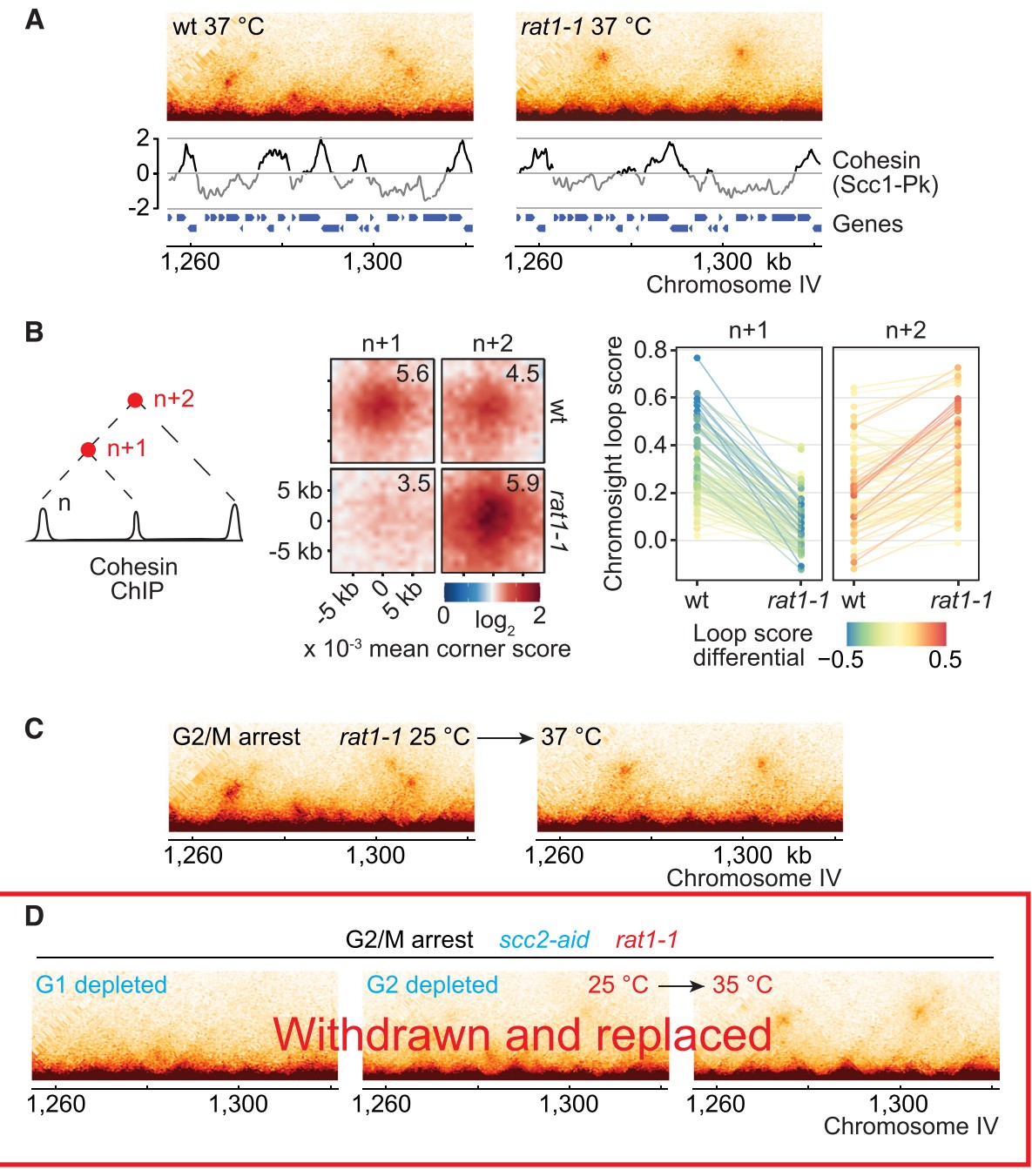

**Figure 2. Transcription directs loop expansion, without the cohesin loader.**

(A) 500 bp-resolution merged micro-C contact maps from two independent experiments of wt and *rat1-1* cells arrested in G2/M at a restrictive temperature (37 °C) for the *rat1-1* allele. Cohesin (Scc1-Pk$_9$) ChIP microarray traces under the same conditions (Ocampo-Hafalla et al, 2016) are shown. (B) Scheme for how genomic regions were selected for analysis, aggregate loop profiles with mean corner scores, and *rat1-1* dependent change of chromosight loop scores at each position ($n = 104$). (C) The experiment in (A) was repeated, but *rat1-1* cells were arrested in G2/M at a permissive temperature (25 °C, left), before temperature shift to 37 °C (right). (D) The experiment in (C) was repeated with a *rat1-1* strain from which the Scc2 cohesin loader subunit could be depleted. Scc2 was either depleted in G1 before release and arrest in G2/M (left), or following arrest in G2/M. Following G2/M depletion, samples were analysed before 25 °C (middle) and after *rat1-1* inactivation by temperature shift to 35 °C (right).

Overall chromosomal cohesin levels were slightly reduced following transcription inhibition (Fig. 3B). When we specifically interrogated cohesin occupancy at loop anchors, we found a less than twofold reduction, in both G1 and G2/M cells, while loop corner scores declined far more steeply (Fig. EV5D). These observations suggest that transcription promotes chromatin loop formation by a mechanism additional to recruiting and positioning cohesin.

The idea that transcription establishes chromatin architecture by blocking loop extrusion stems from the observation of long-range

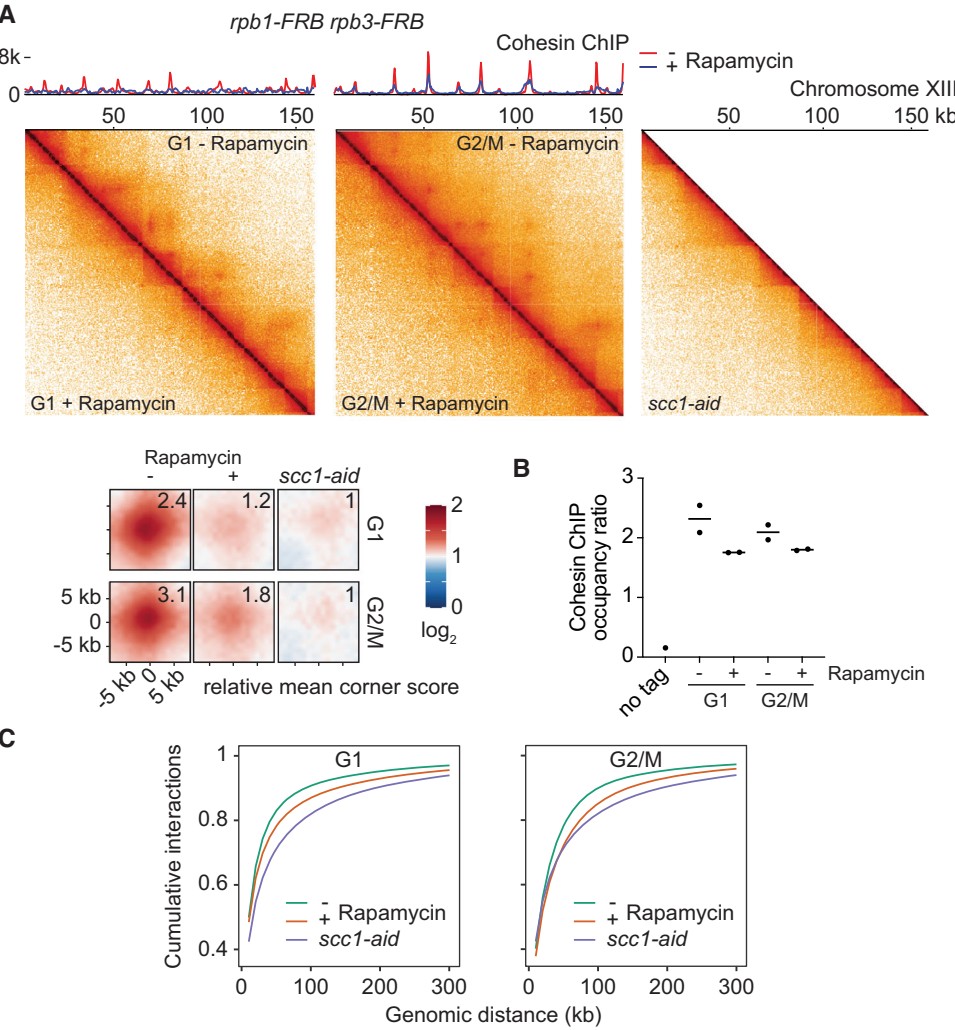

**Figure 3. Transcription promotes cohesin loop formation.**

(**A**) RNA Polymerase II depletion by anchor away. 500 bp-resolution merged micro-C contact maps from two independent experiments of cells arrested in late G1 or G2/M in the absence or presence of rapamycin to deplete RNA polymerase II. Calibrated cohesin (Smc3-Pk$_3$) ChIP-seq traces from the same samples are shown. A merged micro-C contact map of two independent repeats of cohesin-depleted (*scc1-aid*) G2/M cells is shown for comparison. Aggregate chromatin structure of loops detected by chromosight and linked to cohesin anchors in the rapamycin samples ($n = 788$ G1; $n = 1060$ G2/M) are shown, together with mean corner scores relative to those recorded in the *scc1-aid* sample. (**B**) Overall cohesin ChIP occupancy in the two repeat experiments, relative to a *C. glabrata* spike-in. (**C**) Cumulative interaction counts as a function of genomic distance in the above micro-C experiments. Source data are available online for this figure.

intra-chromosome contacts following transcription inhibition—i.e. cohesins would extrude longer loops if unobstructed by RNA polymerases (Banigan et al, 2023; Jeppsson et al, 2022). On the other hand, contact frequency distributions are normalised to unity, and an apparent increase of long-range interactions could equally be the consequence of fewer short-range interactions. To explore whether long-range chromatin contacts after transcription inhibition are cohesin-mediated, we compared contact distributions following transcription inhibition with those following cohesin depletion. As (often-used) logarithmic contact frequency plots place undue emphasis on infrequent long-range contacts, we plotted interactions cumulatively using linear distance and frequency scales (Fig. 3C). This analysis confirmed increased long-range interactions following transcription inhibition but also revealed an even more pronounced

shift toward long-range interactions after cohesin depletion. Thus, long-range interactions occur independently of cohesin, and loss of cohesin-mediated short-range contacts is a probable explanation for the relative shift towards long-range interactions following transcription inhibition.

As contact frequency distributions cannot ultimately decide whether transcription promotes short-range interactions, or limits long-range interactions, we return to considering our earlier experiment in Fig. 2. If transcription limits loop extrusion, then pervasive transcription following *rat1-1* inactivation should have imposed additional constraints on loop extrusion and resulted in smaller loops. On the contrary, transcriptional readthrough resulted in larger loops, suggesting that transcription promotes chromatin loop formation.

## Unwound DNA promotes loop formation

A feature of transcription is unwound DNA, which cohesin might target as a loop-capture substrate akin to cohesion establishment at DNA replication forks. Transcription results in accumulation of positive DNA helical tension ahead of the RNA polymerase, while negative helical tension accumulates behind (Fernández et al, 2014). The single-stranded DNA-binding protein RPA is detectable at sites of transcription (Sikorski et al, 2011), suggesting that negative helical tension results in DNA unwinding. We therefore explored a possible contribution of unwound DNA to chromatin loop formation.

The general transcription coactivator Sub1, a homologue of human PC4, promotes transcription initiation and elongation and contains a C-terminal DNA-binding domain with affinity for unwound DNA (Sikorski et al, 2011; Werten et al, 1998). To test whether Sub1 stabilises open DNA structures that cohesin recognises, we compared the chromatin loop pattern between a wild type and a *sub1Δ* strain, however, could not discern any differences (Fig. 4A; Appendix Fig. S1). Sub1 might perform its roles mainly via its protein interactions with the transcription elongation factor Spt5 (García et al, 2012).

A recent study reported a role of the Chl1 helicase in promoting cohesin loading at unwound DNA substrates (Murayama et al, 2024). Again, the chromatin loop pattern appeared unchanged between a wild-type and a *chl1Δ* strain (Fig. 4B). Chl1 engages in a direct protein interaction with the replisome component Ctf4 (Samora et al, 2016), which might restrict Chl1 action to the context of replication forks.

Finally, we directly modulated DNA helical tension. If unwound DNA is cohesin's loop substrate, then increasing positive helical tension should attenuate loop formation. To increase positive helical tension, we used a strain background in which the abundant endogenous yeast topoisomerases I and II can be inactivated (*top1Δ top2-4*), and *E. coli TopA* is ectopically expressed. *TopA* is a topoisomerase I enzyme that only operates on underwound DNA and thereby causes a net accumulation of positive helical tension (Joshi et al, 2010). *TopA* expression, compared to the *top1Δ top2-4* control, resulted in reduced loop intensities (Fig. 4C). While *TopA* expression might have affected cell physiology beyond modulating DNA topology at transcription sites, these results are consistent with the possibility that unwound DNA forms the cohesin capture target during chromatin loop formation.

## TAD formation without loop extrusion?

In addition to loops, TADs are a prominent cohesin-dependent feature of the mammalian interphase chromatin landscape (Rao et al, 2017; Schwarzer et al, 2017). The loop extrusion hypothesis posits that TADs arise as cohesins bring distal DNA segments into proximity while moving along the chromatin chain (Davidson and Peters, 2021; Higashi and Uhlmann, 2022; Kim et al, 2023a; Yatskevich et al, 2019). On the other hand, oligopaint approaches revealed that TADs form independently of cohesin in individual human cells (Bintu et al, 2018), but that cohesin defines boundaries that make TADs visible to population-based techniques. Above, cohesin depletion in G1 arrested budding yeast cells caused chromatin loop loss, while TAD structures persisted (Fig. 3A). Other chromatin interactions, possibly including depletion

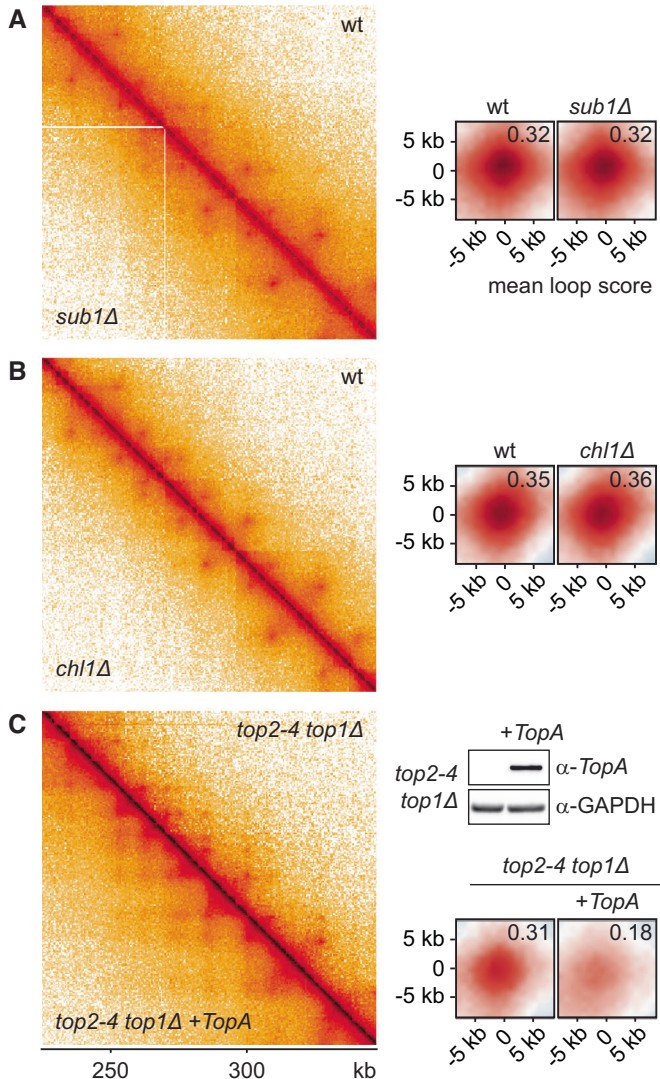

### Figure 4. Unwound DNA as possible cohesin target for loop capture.

(A) 500 bp-resolution contact maps of G2/M arrested wild-type (wt) and *sub1Δ* cells, and comparison of aggregated loops, identified in the wild-type map and linked to cohesin anchors. Mean chromosight loop scores are used to quantify loop intensity, as corner scores did not reliably assess loops with altered shapes, such as those observed below in (C). (B) as (A) but an aggregate map of two independent repeats comparing wild-type and *chl1Δ* cells, arrested in late G1 using Sic1 overexpression, is shown. (C) As (A), but aggregate maps are shown from three independent repeat experiments of *top1Δ top2-4* and *top1Δ top2-4* +*TopA* cells following shift to a restrictive temperature for the *top2-4* allele. *TopA* was detected with an α-*TopA* antibody (Zhou et al, 2017), GAPDH, detected by α-GAPDH antibody (Abcam, clone GA1R, ab125247) served as the loading control. See Appendix Fig. S1 for experimental design and cell synchronisation details of these experiments. Source data are available online for this figure.

attraction (Marenduzzo et al, 2006), apparently suffice to maintain cohesin-independent TADs.

To conclude, we investigated cohesin's contribution to de novo TAD formation. We turned to the budding yeast *GAL7-GAL10-GAL1* locus, encompassing a cluster of three galactose-inducible

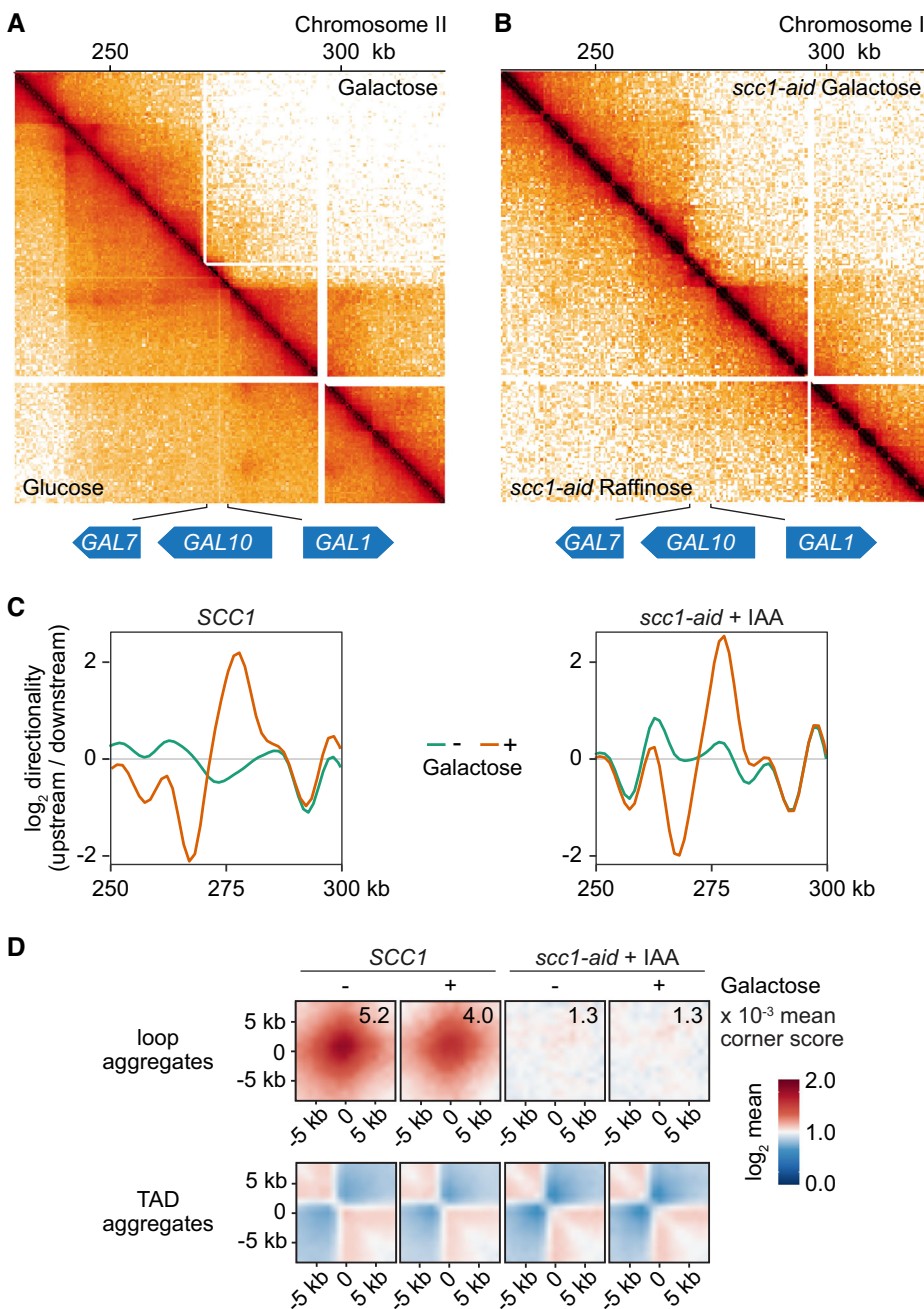

**Figure 5. TAD formation without cohesin.**

(A) In all, 500 bp-resolution merged micro-C contact maps from two independent experiments surrounding the *GAL7-GAL10-GAL1* locus, indicated by a black bracket below the maps, of cells grown in medium with glucose, or with raffinose + galactose (Galactose) as the carbon source. The control cells grown in glucose were also previously analysed as part of Fig. 1C. (B) As (A) but cells were grown in medium containing raffinose as the carbon source. The cohesin subunit Scc1 was depleted by an auxin-inducible degron. Samples were then analysed after further incubation in raffinose medium, or following galactose addition. (C) Interaction directionality plot across the *GAL7-GAL10-GAL1* locus to characterise insulation at the generated domain boundaries. (D) Aggregate loops (including mean corner scores) and TAD boundaries identified genome-wide in the *SCC1* glucose sample and recorded under the indicated conditions.

genes. Cells grown in the presence of glucose, when cluster expression is repressed, show inconspicuous chromatin features at this locus. In contrast, a pronounced TAD boundary is observed in cells grown in galactose-containing medium when cluster expression is switched on (Fig. 5A). Next, we depleted cohesin (Appendix Fig. S2) and only then added galactose to induce cluster expression. As expected, chromatin loop signals were no longer observed in cohesin's absence. Yet, prominent TAD boundaries formed at the *GAL7-GAL10-GAL1* locus, insulating the *GAL* gene cluster from upstream and downstream sequences (Fig. 5B). An interaction

directionality plot confirmed that a TAD boundary of equal strength formed with or without cohesin (Fig. 5C). We also quantified insulation at all TAD boundaries that are detectable in control cells. Insulation remained unchanged following cohesin depletion (Figs. 5D and EV5E). In contrast, quantification of all detectable chromatin loops confirmed a dramatic loop loss. Cohesin-independent TAD formation at the *GAL7-GAL10-GAL1* locus was recently independently observed (preprint: Chapard et al, 2023). The insulating property of a strongly expressed gene cluster, also seen in bacteria (Le and Laub, 2016), appears sufficient to enact a domain boundary.

# Discussion

The loop extrusion hypothesis has shaped current thinking about chromosomal processes. The concept is based on in vitro observations but had not yet been experimentally tested in vivo. We have now studied cohesin variants that lost their in vitro loop extrusion ability, and we have removed a key component of the loop extrusion mechanism, the cohesin loader. In both cases, phenomena ascribed to loop extrusion, the formation and growth of chromatin loops, remained largely unaltered. While in vitro loop extrusion constitutes a striking phenomenon, which under conditions that favour DNA bending might arise as a by-product of the topological cohesin loading reaction (Higashi et al, 2020; Higashi et al, 2021), our observations suggest that in vivo chromatin loops form by a different mechanism.

Instead of loop extrusion, we find that transcription promotes loop formation and acts as an extrinsic motor that expands chromatin loops. While we performed our study using the simple budding yeast model, evidence for transcription-coupled loop formation is apparent in higher eukaryotes. Cohesin promotes sister chromatid cohesion during early vertebrate development, but interphase chromatin structure emerges only once transcription commences during zygotic genome activation (Wike et al, 2021). Loops and TADs are lost every time human cohesin dissociates from chromatin during cell divisions, and their re-establishment depends on transcription (Zhang et al, 2021). Combined single-cell Hi-C and RNA-sequencing revealed that most chromatin interaction changes during developmental transitions coincide with, or follow, transcription changes (Liu et al, 2023), with the exception of enhancer-promoter interactions that precede transcription changes but are established in a cohesin-independent, stochastic search pattern (Aljahani et al, 2022; Brückner et al, 2023; Hsieh et al, 2022). Other studies observed only smaller contributions of transcription to loop formation (Vian et al, 2018; Wike et al, 2021; Zhang et al, 2023), though we note the experimental challenge of efficiently inhibiting transcription. Those studies that reported a transcription requirement for loop formation used increased inhibitor concentrations or combined more than one tool to downregulate transcription (Jeppsson et al, 2022; Zhang et al, 2021).

How might transcription promote chromatin loop formation? Transcribed regions bear resemblance to replication forks – unwound DNA strands in the wake of the RNA polymerase are accessible to ssDNA binding proteins, and probably to cohesin (Sikorski et al, 2011). We therefore propose a unified model for how cohesin establishes DNA–DNA interactions both at DNA replications forks, as well as in regions of active transcription.

Cohesin is at first loaded onto chromosomes at accessible DNA regions, notably nucleosome-free promoter regions (Kagey et al, 2010; preprint Kim et al, 2023b; Lopez-Serra et al, 2014; Mattingly et al, 2022; Muñoz et al, 2019). Cohesin's primary function is to await DNA replication, when it is transferred behind the replisome to engage in cohesion establishment (Lengronne et al, 2006; Murayama et al, 2018). This reaction includes sequential topological entrapment of dsDNA, then ssDNA, on the leading and lagging strands (Fig. 6, left). If the above model for sister chromatid cohesion establishment is correct, it will be inevitable that cohesin, loaded onto dsDNA at promoters, entraps accessible single-stranded regions as transcription commences. (Alternatively, cohesin might entrap dsDNA ahead of the unwound transcription bubble, a substrate geometry for which it displays pronounced affinity (Murayama et al, 2024)). Second DNA capture is followed by subsequent translocation along transcribed units, pushed by the transcription machinery (Fig. 6, right).

Supporting the above model, chromatin contact stripes along active mouse genes are suggestive of cohesin-mediated interactions between open promoter elements and the moving transcription machinery (Vian et al, 2018). In nematodes, similar stripes have been described as 'jets' that stretch out from cohesin loader binding sites at promoters (preprint: Kim et al, 2023b). Stripes that extend from cohesin loading sites in the direction of transcription are easier explained by an active role of transcription, rather than by a barrier function. If cohesin is stable enough, e.g. following downregulation of its release factor Wapl, loop expansion could extend over long distances, e.g. in the case of locus scanning during V(D)J or class switch recombination (Dai et al, 2021).

A mechanism for loop formation in which loop anchors find each other by Brownian diffusion, as proposed here, suggests that cohesin is not limited to establishing short-range interactions. With a lower frequency, cohesin would also capture longer-range loops, consistent with the simultaneous establishment of both short- and long-range loops following cohesin depletion and re-addition in human cells (Rao et al, 2017). A transcription-aided loop-capture mechanism can also explain the generation of both stem- and circle-loops in the fruit fly, the latter of which cannot form by loop extrusion (preprint: Bing et al, 2024). A capture mechanism furthermore makes the strong prediction that cohesin is not restricted to intra-chromosome interactions but, at least occasionally, establishes contacts between chromosomes. The chromatin interaction spectrum of fission yeast cohesin, like that of the related condensin complex, indeed includes both intra- and inter-chromosome associations (Kim et al, 2016; Tang et al, 2023).

If life without loop extrusion sounds provocative, our results reveal that in vitro loop extrusion diverges from in vivo chromatin loop formation. In its place, transcription events occupy roles in promoting loop formation and expansion. A limit of our study is that we compare in vitro loop extrusion behaviour to in vivo chromatin loop formation, however this limitation is not unique to our study. A challenge for the future lies in observing in vivo cohesin behaviour in real time and at sufficient resolution to distinguish loop-capture and loop extrusion events. For now, our findings highlight the gap between in vitro single-molecule observations and in vivo chromatin behaviour. They motivate a rethinking of how the chromosomal cohesin complex shapes the interphase genome.

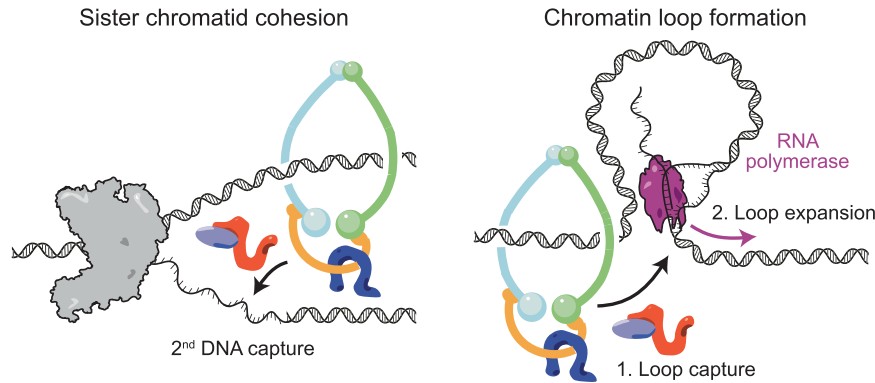

**Figure 6.  A unified model for sister chromatid cohesion establishment and chromatin loop formation by the cohesin complex.**

Cohesin sequentially and topologically entraps two DNAs (Murayama et al, 2018; Richeldi et al, 2024). While the prime purpose of this reaction is the establishment of sister chromatid cohesion at DNA replication forks, second DNA capture by the same mechanism occurs at sites of ongoing transcription, resulting in transcription-dependent interphase chromatin domain architecture.

# Methods

## Yeast culture

All yeast strains used in this study were of W303 background and are listed in Appendix Table S1. *SCC3*[3E]- and *SMC1*[4E]-strains were generated by altering the endogenous gene loci using gene targeting constructs. Successful targeting was confirmed by PCR-based genotyping and DNA sequencing. Cells were grown at 30 °C in YPD medium, except if stated otherwise. Asynchronous mid-log phase cells were diluted to an optical density of $OD_{600} = 0.2$. Cell synchronisation in G1 was achieved by the addition of 7.5 μg/ml of the mating pheromone α-factor, every hour, for two hours. For late G1 arrest, after cohesin subunit Scc1 expression commences but before the onset of DNA replication (Lopez-Serra et al, 2013), cells were grown in YP medium containing 2% raffinose as the carbon source. Sic1[V5,V33,A76] expression was induced by 2% galactose addition at the same time as α-factor addition, and the arrest was extended to three hours. Cells were then washed by filtration and released in raffinose and galactose-containing medium without α-factor. G2/M arrest was achieved by releasing cells from α-factor synchronisation into YPD medium containing 6 μg/ml nocodazole. For RNA polymerase II anchor away, 1 μg/ml rapamycin was added for 2 h before analysis. Scc2 or Scc3 depletion were achieved by addition of 2 mM methionine to shut off *MET3* promoter expression, as well as 1 mM auxin (indole-3-acetic acid). For experiments with the *rat1-1* strain, cells were grown at 25 °C and, as indicated, the temperature was shifted to 37 °C in a water bath, or gradually raised to 35 °C in an air incubator. *top1Δ top2-4* cells, with or without the *TopA* vector, were cultivated in medium lacking leucine before shifting to YPD medium for five population doublings and temperature shift to 35 °C for 2 h.

## Cell cycle profiling by flow cytometry

Cells were harvested and fixed in 70% ice-cold ethanol for at least 2 h before pelleting and resuspension in 50 mM Tris-HCl pH 7.5 including 0.1 mg/ml RNase A and incubation overnight at 37 °C. DNA was stained with 25 μg/ml propidium iodide in 200 mM Tris-HCl pH 7.5, 210 mM NaCl, 78 mM MgCl$_2$. Cells were sonicated and diluted in 50 mM Tris-HCl pH 7.5. Flow cytometric analyses were performed using an LSRFortessa X-20 flow cytometer (BD Biosciences). In total, 10,000 cells were counted for each sample. Results were visualised using FlowJo.

## Cell viability assays

Asynchronously growing yeast cultures were adjusted to equal optical densities, then 3 μl drops of tenfold serial dilutions were applied to YPD agar plates, to which indicated supplemental chemicals had been added at the given concentrations. Plates were incubated at 30 °C degrees for 2 days.

## Sister chromatid cohesion assay

Strains expressing a tetR-GFP fusion protein and harbouring tetO repeats integrated at the *URA3* locus (Michaelis et al, 1997) were synchronised by α-factor treatment and released for 2 h into YPD medium containing 6 μg/ml nocodazole. Culture aliquots were harvested and fixed in 70% cold ethanol overnight. Cells were resuspended in PBSA, sonicated, and mounted on 2% agarose patches prepared on glass slides. Z-stacks of 20 images at 0.25-μm intervals were acquired using a DeltaVision Olympus IX70 inverted microscope equipped with a 100× (NA = 1.40) PlanApo objective, deconvolved, and merged using maximum intensity projection. The numbers of visually discernible GFP dots in each cell were then manually counted. More than one GFP dot was taken to indicate defective sister chromatid cohesion. One hundred cells were counted for each strain, experiments were three times independently repeated.

## Calibrated ChIP-sequencing analysis

In all, 50 $OD_{600}$ units of *Saccharomyces cerevisiae* cells were mixed at a 5:1 ratio with *Candida glabrata* cells in which the Smc3 subunit was fused to a Pk epitope tag (Hu et al, 2015), and broken in 700 μl Lysis Buffer (50 mM HEPES-KOH pH 7.5, 140 mM NaCl, 1 mM EDTA, 1% Triton, 0.1% sodium deoxycholate, supplemented with

protease inhibitors) by glass bead rupture. The lysate was retrieved by centrifugation for 1 min at 2000 rpm and the supernatant then removed after centrifugation for 10 min at maximum speed in a benchtop centrifuge. The pellet was resuspended in 1 ml lysis buffer and sonicated to reach an average DNA fragment size of 200–300 bp. Cohesin was precipitated using an α-Pk antibody (Bio-Rad, RRID: AB_322378) and protein A-coupled Dynabeads for 2 h. Beads were washed three times with Lysis Buffer, three times with Lysis Buffer supplemented with 500 mM NaCl and two times with Wash Buffer (10 mM Tris-HCl pH 8.0, 250 mM LiCl, 0.5% NP-40, 0.5% sodium deoxycholate). The samples were resuspended in 10 mM Tris-HCl pH 8.0, 1 mM EDTA, 1% SDS (including 3 µg/ml proteinase K and 2 µg/ml RNase A) and de-cross-linked overnight at 65 °C. DNA was purified by phenol-chloroform extraction and the aqueous fraction was further cleaned using DNA Clean & Concentrator-5 with Zymo-Spin Columns (Zymo Research). Libraries were built with the NEBNext Ultra II DNA Library Prep Kit for Illumina (New England Biolabs) following the manufacturer's instructions. Sequencing was performed on an Illumina NovaSeq 6000 sequencer.

Sequence data analysis was performed using the nf-core/chipseq pipeline (https://doi.org/10.5281/zenodo.3240506; Ewels et al, 2020). Sequences were aligned to the W303 genome (https://www.ncbi.nlm.nih.gov/nuccore/LYZE00000000) (Matheson et al, 2017). Calibration was then performed as described (Hu et al, 2015). ChIP-seq traces were plotted using PyGenomeTracks (3.8) (Lopez-Delisle et al, 2021). Cohesin peaks were called using MACS2 (2.2.9.1) (Zhang et al, 2008) in paired mode with $q < 0.01$ as the threshold.

## ChIP microarray analysis

ChIP microarray analysis was performed using Affymetrix GeneChip Yeast Genome 2.0 arrays as described (Liu et al, 2020). Output files were then transposed from *S. cerevisiae* S288C to W303 genome annotations using a liftOver chain (http://hgdownload.soe.ucsc.edu/admin/exe/).

## Micro-C analysis

Micro-C was performed following published protocols (Costantino et al, 2020; Slobadyanyuk et al, 2022) with minor modifications. Yeast cultures were cross-linked with 3% formaldehyde for 15 min at 30 °C. The reactions were quenched by the addition of 250 mM glycine at 30 °C for 5 min with agitation. Cells were pelleted by centrifugation at 4000 rpm at 4 °C for 5 min and washed twice with water. Cells were then resuspended in Buffer Z (50 mM Tris-HCl pH 7.5, 1 M sorbitol, 10 mM β-mercaptoethanol) and spheroplasted by addition of 250 µg/ml Zymolyase (100 T) in a 30 °C incubator at 200 rpm for 40–60 min. Spheroplasts were washed once by 4 °C PBS and then pelleted at 4000 rpm at 4 °C for 10 min. Chromatin was further cross-linked by suspending pellets in PBS supplemented with 3 mM disuccinimidyl glutarate (ThermoFisher) and incubated at 30 °C for 40 min with gentle shaking before quenching by addition of 400 mM glycine for 5 min at 30 °C. Cells were pelleted by centrifugation at 4000 rpm at 4 °C for 10 min, washed once with ice-cold PBS and stored at −80 °C. Pellets were then treated as described (Costantino et al, 2020) until the de-crosslinking step. De-crosslinking solution was added with an

equal volume of phenol-chloroform-isoamyl alcohol (25:24:1), vortexed intensively and centrifuged for 15 min at room temperature. The aqueous phase was purified using a ZymoClean column (Zymo Research) according to the manufacturer's instructions. Dinucleosome-sized DNA fragments were purified and excised from a 3% NuSieve GTG agarose gel (Lonza) using the Zymoclean Gel DNA Recovery Kit (Zymo Research). Micro-C libraries were then prepared using the NEBNext Ultra II DNA Library Prep Kit for Illumina (New England Biolabs) as described (Slobadyanyuk et al, 2022) and sequenced using an Illumina NovaSeq 6000.

Micro-C datasets were processed through the Distiller pipeline (https://github.com/open2c/distiller-nf, commit 8aa86e) to implement read filtering, alignment, PCR duplicate removal, binning and balancing of replicate sample matrices. Reads were aligned to the W303 genome using bwa (0.17.7), and the resulting maps filtered to remove low-quality alignments (MAPQ < 30) and cis alignment pairs within 150 bp. Replicates were analysed independently, and their quality was assessed before aggregation into sample-level datasets. Read count statistics can be found in Appendix Table S2. Each presented map is the aggregate of two replicates, except Fig. 4B which is based on a single experiment and Fig. 4C which is the aggregate of three repeats. Reproducibility between replicates was verified by Spearman correlation and by comparing an individual to aggregate loop scores (Appendix Fig. S3). Maps were visualised and explored using HiGlass (Kerpedjiev et al, 2018). Loops were called using chromosight (https://github.com/koszullab/chromosight; Matthey-Doret et al, 2020), as described (Kakui et al, 2022), using a threshold ≥0.3, and intersected with cohesin ChIP-seq peaks. Aggregate profiles were generated using chromosight, with datasets subsampled to the lowest-depth dataset in the comparison. Cooler files were read into R (4.1.1) using HiCExperiment (1.0.0) (https://doi.org/10.18129/B9.bioc.HiCExperiment) and loop files read using InteractionSet (1.22.0) (Lun et al, 2016). To compare loop intensities, we applied the HiCExperiment corner score metric that measures the interaction differential between the centre and corner of each loop region. To plot directionality scores, neighbouring bins at 5 kb resolution were considered across chromosome II coordinates 250 - 300 kb. For each bin, the sum of balanced interactions within 50 kb upstream or downstream were calculated. The $\log_2$ ratio of upstream/downstream was plotted and smoothed using geom_smooth with span = 0.25 from the ggplot2 package. Hi-C matrices in mcool files were accessed using HiContacts (1.2.0).

## Cohesin and cohesin loader purification

Cells harbouring Smc1, Smc3, Scc1, and Scc3 expression vectors were grown in YP medium containing 2% raffinose to $OD_{600} = 1.0$ at 30 °C. 2% galactose was added to the culture to induce protein expression for 2 h. For purification of Scc3$^{3E}$- and Smc1$^{4E}$-cohesin, both the overexpressed and endogenous copies of these subunits carried the respective mutations (Scc3 K423E, K520E, K669E and Smc1 R53E, R58E, N60E, K63E). Cells were collected by centrifugation, washed once with PBSA and resuspended in an equal volume of Lysis Buffer (50 mM Tris-HCl pH 7.5, 300 mM NaCl, 1 mM MgCl$_2$, 10% glycerol, 0.5 mM TCEP, 0.5 mM Pefabloc (Roche) and an additional protease inhibitor cocktail (cOmplete, Roche)). The cell suspension was frozen in liquid nitrogen and broken in a freezer mill. The cell powder was thawed on ice and adjusted with two volumes of Lysis Buffer complemented with 50

U/ml benzonase. The lysates were clarified by centrifugation at 30,000 × g for 15 min at 4 °C, then at 142,000 × g for 1 h. The clarified lysate was passed through a HiTrap NHS-Activated HP affinity column adsorbed with rabbit IgG according to the manufacturer instructions (Cytiva) and washed with Lysis Buffer, Lysis Buffer including 1 mM ATP, and then Lysis buffer including 0.01% NP-40, before overnight incubation with 10 µg/ml PreScission protease. The eluate was loaded onto a HiTrap Heparin HP column (Cytiva) that was developed with a linear gradient from 300 mM to 1 M NaCl in buffer A (50 mM Tris-HCl pH 7.5, 10% glycerol, 0.5 mM TCEP). The peak fractions were pooled and loaded onto a Superose 6 10/300 GL gel filtration column (Cytiva) that was equilibrated and developed with Buffer A containing 150 mM NaCl. The peak fractions were concentrated by ultrafiltration. The Scc2–Scc4 cohesin loader complex was purified as described (Minamino et al, 2018).

## Electrophoretic gel mobility shift assay

Increasing concentrations of cohesin were incubated for 30 min with 16.6 nM (molecules) 505 bp dsDNA (a PCR product from yeast genomic DNA) at 30 °C in 40 mM Tris-HCl pH 7.5, 50 mM NaCl, 2 mM MgCl$_2$, 0.5 mM ATP, 0.5 mM TCEP. The reactions were then separated by 0.8% agarose/TAE gel electrophoresis. DNA was detected by staining with SYBR Gold for an hour, followed by 15 min destaining in TAE before imaging with a Gel Doc XR+ (Bio-Rad) gel documentation system.

## In vitro cohesin loading assay

In a reaction volume of 15 µl, 30 nM cohesin, 60 nM Scc2–Scc4 cohesin loader and 3.3 nM (molecules) pBluescript II KS( + ) DNA were mixed in 35 mM Tris-HCl pH 7.0, 20 mM NaCl, 0.5 mM MgCl$_2$, 13.3% glycerol, 0.5 mM ATP, 0.003% Tween, and 1 mM TCEP. The reactions were incubated at 30 °C for one hour. Reactions were stopped by the addition of 500 µl of IP Buffer 1 (35 mM Tris-HCl pH 7.5, 100 mM NaCl, 10 mM EDTA, 5% glycerol, 0.35% Triton X-100). In total, 4 µg of α-Pk antibody (clone SV5-Pk1, Bio-Rad) were added and incubated on a wheel at 4 °C for 2 h, before the addition of 40 mg/ml protein A-coupled Dynabeads followed by 30 min additional incubation. The beads were washed three times with IP Buffer 1, twice with IP Buffer 2 (35 mM Tris-HCl pH 7.5, 300 mM NaCl, 10 mM EDTA, 5% glycerol, 0.35% Triton X-100) and once with (35 mM Tris-HCl pH 7.5, 100 mM NaCl, 0.1% Triton X-100). The beads were suspended in 12 µl of elution buffer (10 mM Tris-HCl pH 7.5, 1 mM EDTA, 50 mM NaCl, 0.75% SDS, 1 mg/ml protease K) and incubated at 37 °C for 30 min. The recovered DNA was analysed by 0.8% agarose/TAE gel electrophoresis. The gel was stained with SYBR Gold, as above. Gel images were captured, and band intensities quantified using Fiji (2.3.0/1.53q).

## DNA loop extrusion assays

Microfluidic flow cells were prepared as previously described (Higashi et al, 2021). Flow cells were incubated with 1 µl of α-digoxigenin antibody (Roche, 150 U) diluted in 30 µl TB buffer (40 mM Tris-HCl pH 7.5, 50 mM NaCl) for 10 min and washed with 400 µl TB buffer. The surface of the flow cell was further passivated by incubating with 50 µl of Pluronic F127 (Sigma-

Aldrich, 1% solution in TB buffer) for 10 min followed by washing with 400 µl TB and incubating with 40 µl of β-Casein (Sigma-Aldrich, 10 mg/ml in TB buffer) for 30 min. Subsequently, the flow cell was washed four times with 400 µl TB buffer. 40 µl of 5 pM λ-phage DNA (New England Biolabs), digoxigenin-labelled at both ends (Higashi et al, 2021), was introduced into the flow cell in TB buffer at a flow rate of 4 µl/min using a syringe pump (Harvard Apparatus, Pico Plus Elite 11). The flow cell was then washed again with 40 µl TB buffer at a flow rate of 4 µl/min.

Prior to imaging, the flow cell was equilibrated with 50 µl buffer R (40 mM Tris-HCl pH 7.5, 50 mM NaCl, 2 mM MgCl$_2$, 5 mM ATP, 10 mM DTT, 200 nM SYTOX Orange, 0.2 mg/ml glucose oxidase, 35 µg/ml catalase, 4.5 mg/ml dextrose and 1 mg/ml β-casein) at 15 µl/min. To assess loop extrusion under physiological buffer conditions, the above buffer was modified to contain 100 mM NaCl, 1 mM ATP and 100 nM SYTOX Orange. Cohesin tetramer complex was pre-mixed with its loader at an equimolar ratio at 500 nM concentration in buffer R on ice. DNA loop extrusion was initiated by flowing in the cohesin loader mixture at 5 nM concentration in buffer R at a flow rate of 6 µl/min. DNA molecules stained with SYTOX Orange were imaged using a custom-built HILO microscopy setup utilising a 561 nm laser and a Nikon SR HP Apo TIRF 100x/1.49 oil immersion objective by taking snapshots with 100 ms exposure every second for 8 min. To determine the efficiency of DNA loop extrusion in the absence of the flow, the flow was stopped, and the flow cell was incubated with cohesin, loader and ATP for 8 min. Flow was then resumed, and snapshots taken every second for 1 min. Images were collected with an Andor Sona sCMOS camera, saved as uncompressed TIFF files and further processed using Fiji. Experiments were performed at room temperature.

## Analysis of DNA loop extrusion experiments

To determine the efficiency of DNA loop extrusion, the number of DNA molecules containing loops was divided by the total number of double-tethered DNA molecules. Single-tethered DNA molecules were excluded from the analysis. Loop extrusion rates were extracted as previously described (Higashi et al, 2021). The length of DNA molecules stretched with a constant flow was manually measured before loop extrusion, averaged over a 5 s interval, and normalised to 48.5 kb (the length of a λ-phage DNA). Subsequently, the length of DNA outside the loop was measured, frame by frame, and subtracted from the initial DNA length to calculate the loop size. The loop extrusion rate was calculated as the slope of a linear fit to the measurements.

## Data availability

All unique materials generated in this study are available from the corresponding author without restrictions. The datasets generated and analysed in this study are available in the following databases: (1) Micro-C and ChIP-sequencing data are available from the GEO database under the accession number GSE248282 (2) Cohesin ChIP microarray from (Ocampo-Hafalla et al, 2016) are available from the GEO database under the accession number GSE80464. (3) Rpb1 ChIP-sequencing data from (Baejen et al, 2017) are available from the GEO database under the accession number GSE79222.

The source data of this paper are collected in the following database record: biostudies:S-SCDT-10_1038-S44318-024-00202-5.

## Peer review information

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

## Acknowledgements

The authors would like to thank J Campbell, D Lang and R Mitter for bioinformatics support, C Bouchoux, D Jackson and the Crick Advanced Sequencing Facility for their help, P. Cramer and A. Gammie for sharing data files, J Roca for the *TopA* expression construct, Y-C Tse-Dinh for the *TopA* antibody, D Bentley, L-Y Chu, K Dubrana, Y Kakui, and J Svejstrup for discussions, as well as past and present members of our laboratory for critical reading of the manuscript. This work was supported by the Wellcome Trust (220244/Z/20/Z to FU) and The Francis Crick Institute, which receives its core funding from Cancer Research UK, the UK Medical Research Council, and the Wellcome Trust (cc2125 to MM; cc2137 to FU).

## Author contributions

**Thomas M Guérin**: Investigation; Writing—original draft. **Christopher Barrington**: Investigation; Writing—review and editing. **Georgii Pobegalov**: Investigation; Writing—review and editing. **Maxim I Molodtsov**: Investigation; Writing—review and editing. **Frank Uhlmann**: Investigation; Writing—original draft.

Source data underlying figure panels in this paper may have individual authorship assigned. Where available, figure panel/source data authorship is listed in the following database record: biostudies:S-SCDT-10_1038-S44318-024-00202-5.

## Funding

## Disclosure and competing interests statement

Frank Uhlmann is a member of the Advisory Editorial Board of *The EMBO Journal*. This has no bearing on the editorial consideration of this article for publication.

# Expanded View Figures

**Figure EV1. Characterisation of Scc3³ᴱ and Smc1⁴ᴱ loop extrusion defective cohesin complexes.**

(A) Purified wild type (wt), Scc3³ᴱ-, and Smc1⁴ᴱ-cohesin and cohesin loader were analysed by SDS-PAGE followed by Coomassie Blue staining. (B) Loop extrusion rates, measured as described (Higashi et al, 2021), of wt and Smc1⁴ᴱ-cohesin, in the presence of loader and ATP ($n_{wt} = 37$, $n_{Smc1^{4E}} = 16$). Dashed and dotted lines represent the median and quartile ranges, respectively. Processive extrusion by Smc1⁴ᴱ-cohesin suggests that this variant is defective in loop initiation but less so loop extension. Indeed, Smc1⁴ᴱ-cohesin shows a greater median extrusion rate, which might arise if the small number of loop extrusion events by this variant are biased towards DNAs under low tension on which extrusion proceeds relatively faster. (C) DNA affinity of wt, Scc3³ᴱ- and Smc1⁴ᴱ-cohesin as measured by an electrophoretic mobility shift assay. Increasing cohesin concentrations were between 32 and 525 nM in 2-fold steps. (D) Assay to measure topological (high-salt-resistant) loading of wt, Scc3³ᴱ- and Smc1⁴ᴱ-cohesin onto DNA (Minamino et al, 2018), in the presence of the indicated components. An example agarose gel of the recovered DNA is shown, as well as quantification of the individual results from two independent repeat experiments. Bars show the means. (E) Loop extrusion assay as in Fig. 1C, but the flow cell was incubated with wt, Scc3³ᴱ- or Smc1⁴ᴱ-cohesin, loader and ATP in the absence of flow, before flow was applied to visualise loops. The fractions of DNA with loops were counted in three independent repeat experiments. Individual data points are shown, bars represent the mean and error bars the standard deviation ($n_{wt} = 224$, $n_{Scc3^{3E}} = 269$, $n_{Smc1^{4E}} = 633$). (F) As Fig. 1C in the presence of flow, but a buffer containing 100 mM NaCl was used. See the Methods for complete buffer descriptions. Bars represent the mean and error bars the standard deviation ($n_{wt} = 452$, $n_{Scc3^{3E}} = 295$, $n_{Smc1^{4E}} = 242$).

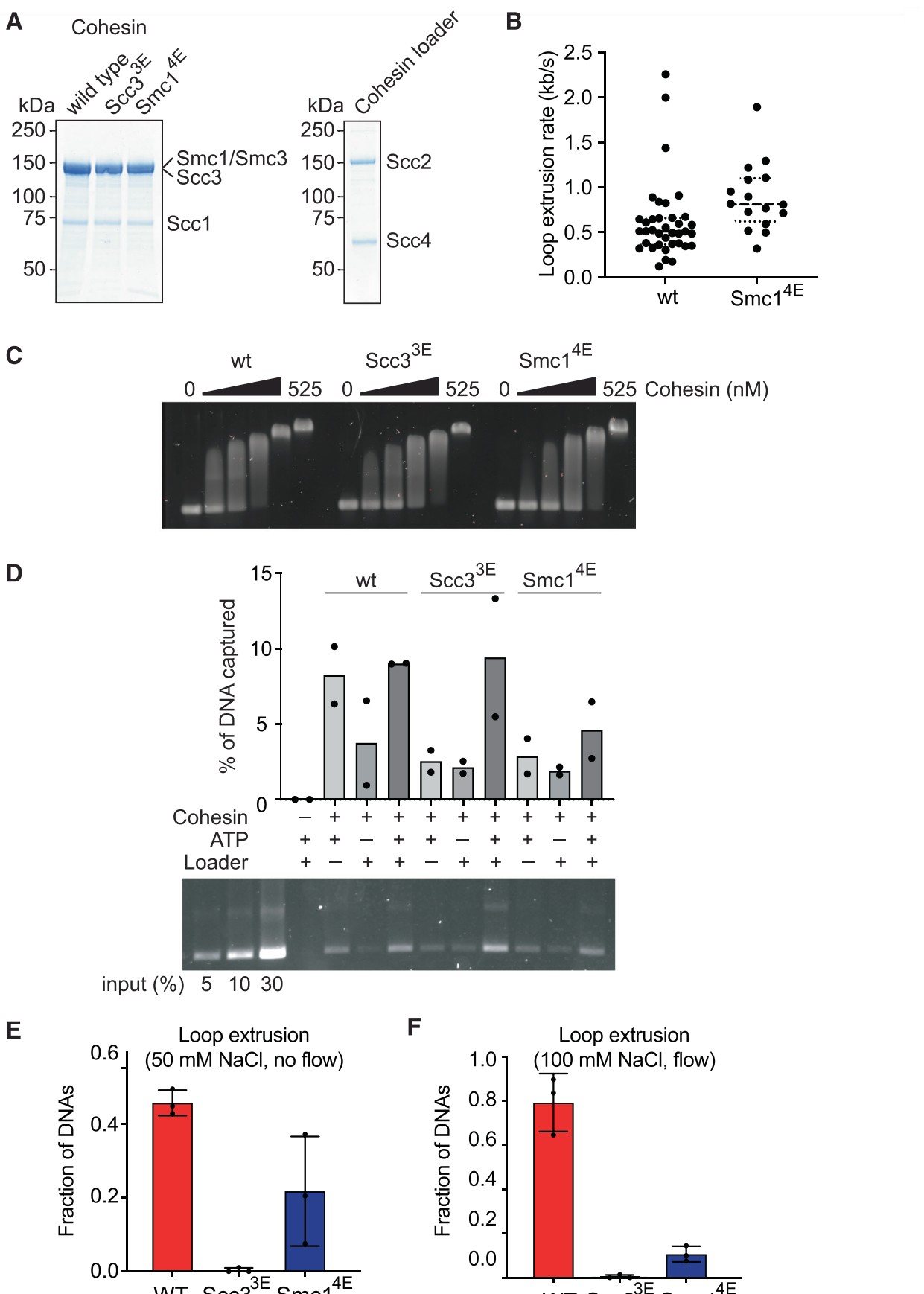

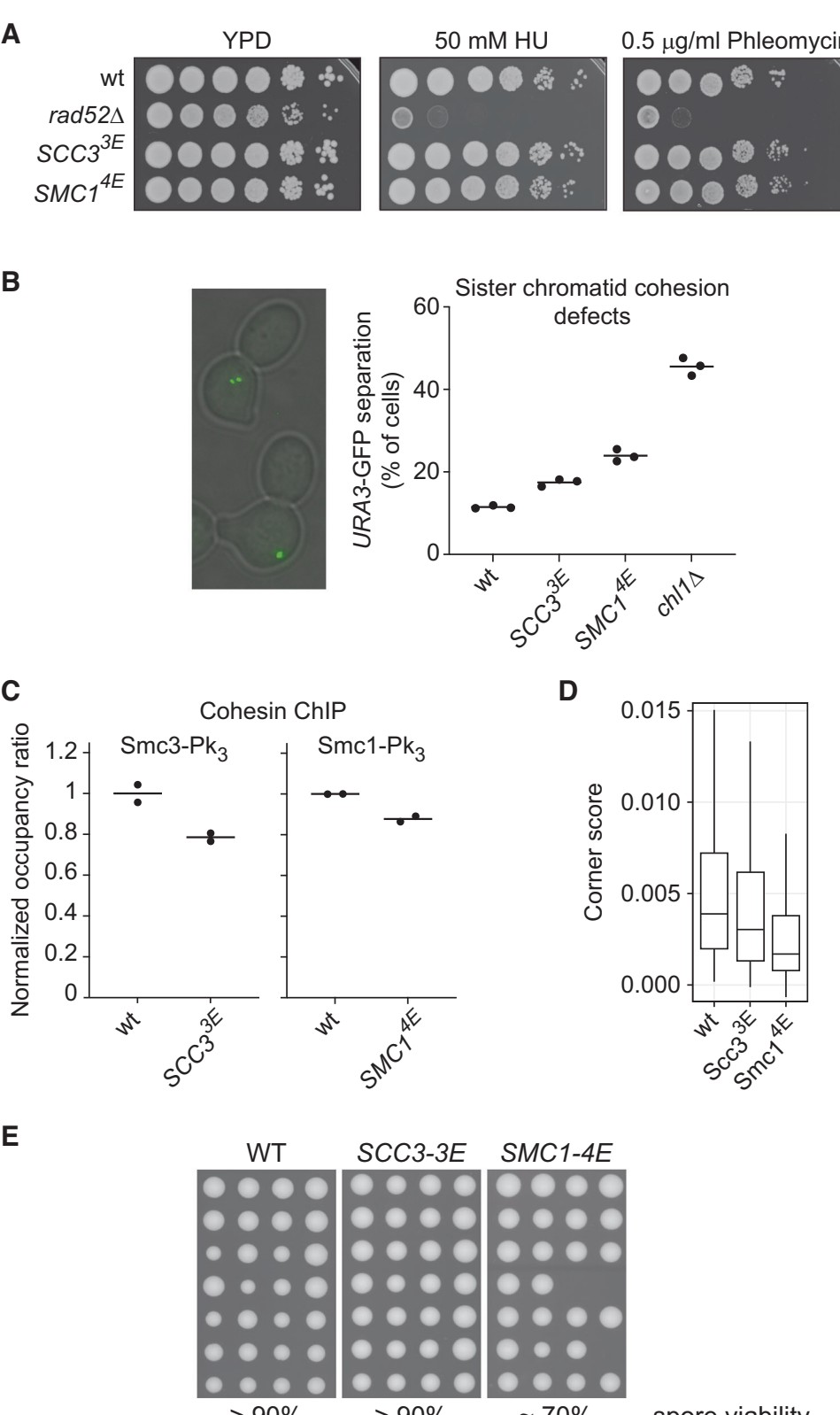

◀

**Figure EV2.   Life without loop extrusion.**

(**A**) 10-fold serial dilutions of cultures of the indicated genotypes were plated onto YPD agar plates containing the indicated compounds and grown at 30 °C for 2 days. A wt and a DNA repair deficient (*rad52Δ*) strain were included as controls. (**B**) Sister chromatid cohesion in G2/M arrested cells was monitored at the GFP-marked *URA3* locus (Michaelis et al, 1997). A representative image of two G2/M arrested cells with intact (left) or defective (right) sister chromatid cohesion is shown. The percentage of cells ($n = 100$) with two separated GFP dots were recorded in three independent repeat experiments. The means are represented by horizontal bars. A wild type (wt) and a cohesion establishment defective (*chl1Δ*; Samora et al, 2016) strain served as controls. (**C**) Overall cohesin ChIP enrichment ratios of wt, compared to Scc3[3E]- and Smc1[4E]-cohesin, relative to a *C. glabrata* spike-in. Cohesin ChIP used Smc3-Pk$_3$ in the Scc3[3E] strain, or Smc1[4E]-Pk$_3$, normalised against Smc3-Pk$_3$ and Smc1-Pk$_3$ wt control strains. (**D**) Corner score distributions of loops identified in the wild-type micro-C contact map and linked to cohesin anchors, sampled in the Scc3[3E]- and Smc1[4E]-maps ($n = 1060$). Box plots represent the median (centre), quartiles (box) and range (whiskers). (**E**) Tetrad dissection following sporulation of homozygous diploid wild type, *SCC3[3E]* and *SMC1[4E]* strains. Spore viability was calculated based on $n = 118/128$, $104/100$ and $82/120$ germinating and colony forming spores, respectively.

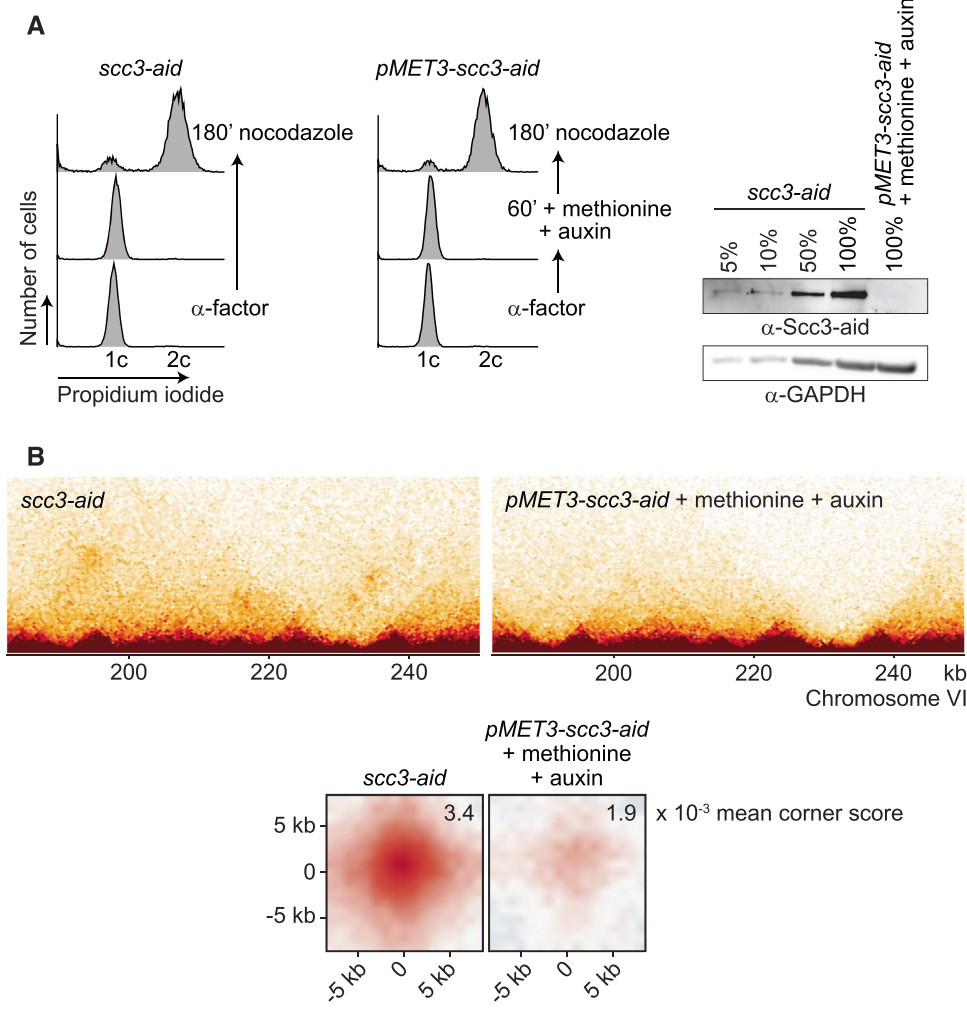

**Figure EV3. Scc3 is required for chromatin loop formation.**

(A) FACS analysis of DNA content, as well as experimental outline, of the experiment to deplete Scc3 by promoter shut-off and an auxin-inducible degron (*pMET-scc3-aid* cells). As a control, we used cells in which *scc3-aid* is expressed under control of its endogenous, methionine-insensitive promoter and to which we added methionine but not auxin during G1 arrest, before release into nocodazole-containing medium for arrest in G2/M. Scc3 depletion was confirmed by Western blotting. Serial dilutions of the control sample without auxin addition were loaded, as well as the depleted sample. Scc3 was detected using an α-aid-tag antibody (Cosmo Bio, CAC-APC004AM). GAPDH, detected by an α-GAPDH antibody (abcam, clone GA1R, ab125247) served as a loading control. (B) 500 bp-resolution merged micro-C contact maps from two independent experiments with Scc3-depleted *pMET-scc3-aid* and control *scc3-aid* cells. Aggregate chromatin loop profiles, detected by chromosight and linked to cohesin anchors in a wild-type strain without any cohesin alteration (Fig. 1C), were recorded in both present maps.

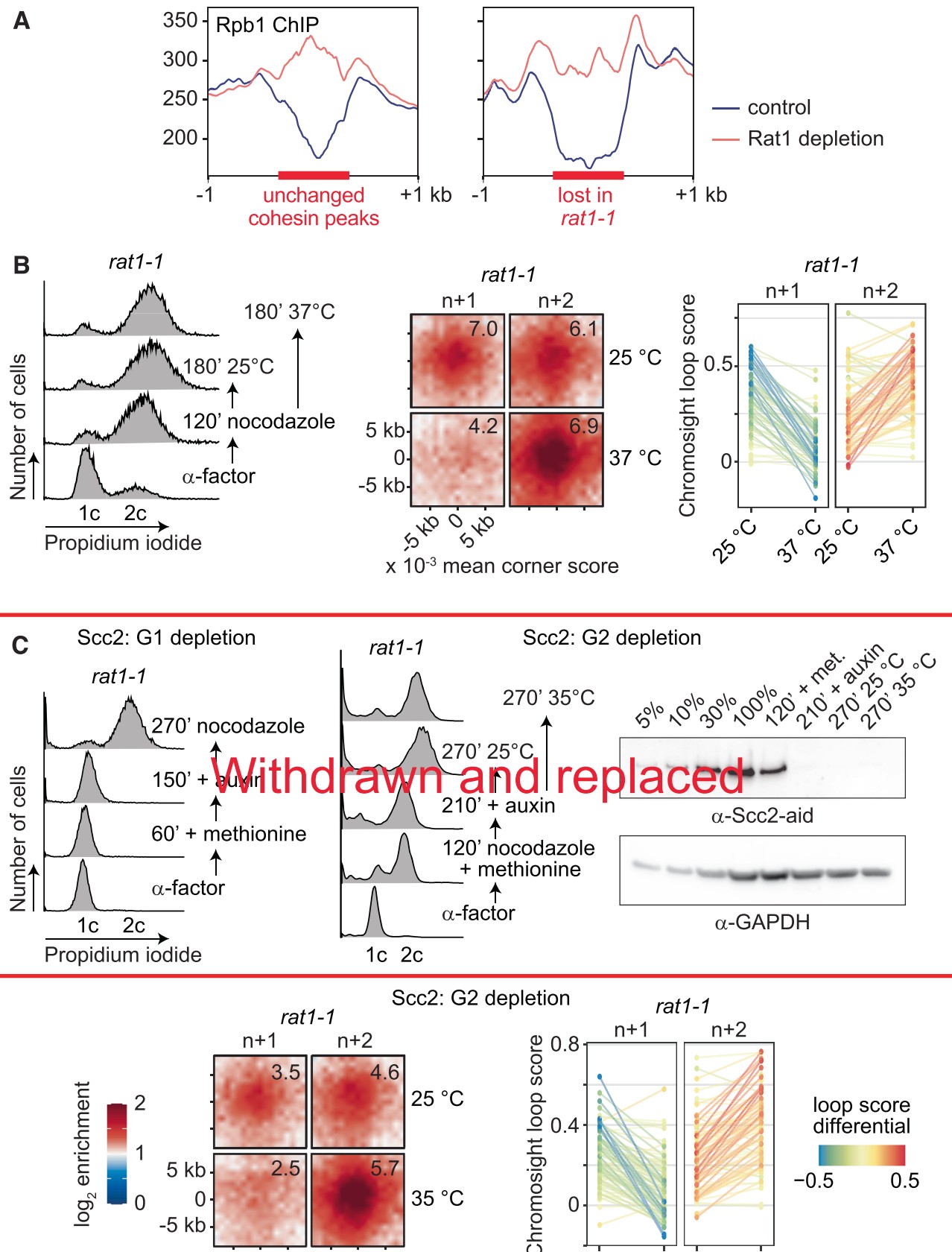

**Figure EV4.  Transcription expands cohesin-mediated chromatin loops.**

(A) Characterisation of cohesin peaks that are displaced following *rat1-1* inactivation. Aggregated Rpb1 ChIP profiles (Baejen et al, 2017) are shown over scaled cohesin peak regions, and their surroundings, that remained either unchanged or that were lost following *rat1-1* inactivation. Before Rat1 depletion (anchor away was used by Baejen et al, 2017), cohesin peaks that will be displaced show strict Rpb1 avoidance. In contrast, cohesin peaks that will remain unchanged were already partly Rpb1 occupied. Following Rat1 depletion, Rpb1 broadly overlapped with both type of regions. We confirmed that differing cohesin peak widths did not cause these differences. To conduct these analyses, raw sequences from (Baejen et al, 2017) were aligned to the S288C genome for analysis using the standard nf core chipseq procedure. Bam files were then converted to BigWigs using bamCoverage with normalizeUsing RPKM and ignoreDuplicates parameters. Binsize was selected at 20 bp and data were smoothed over 3 bins. For comparison we overlaid our previous cohesin (Scc1) ChIP microarray analysis (Ocampo-Hafalla et al, 2016) and selected peaks exclusive to control cells. Peaks longer than 4000 bp or shorter than 500 bp were excluded from the analysis. (B) FACS analysis of DNA content of the cells in the experiment shown in Fig. 2C, together with an experimental outline. Aggregate chromatin profiles of loops ($n = 91$), identified as in Fig. 2B, and a graph depicting the *rat1-1* dependent loop score changes. (C) FACS analyses of DNA content of the cells in the experiment shown in Fig. 2D, together with experimental outlines. Western blot analysis confirmed Scc2 depletion by an auxin-inducible degron. Samples at the indicated times in the experiment are shown. Scc2 was detected using the aid-tag antibody. Tubulin served as a loading control. Aggregate loop profiles ($n = 52$) and a graph depicting the *rat1-1* dependent loop score changes are shown.

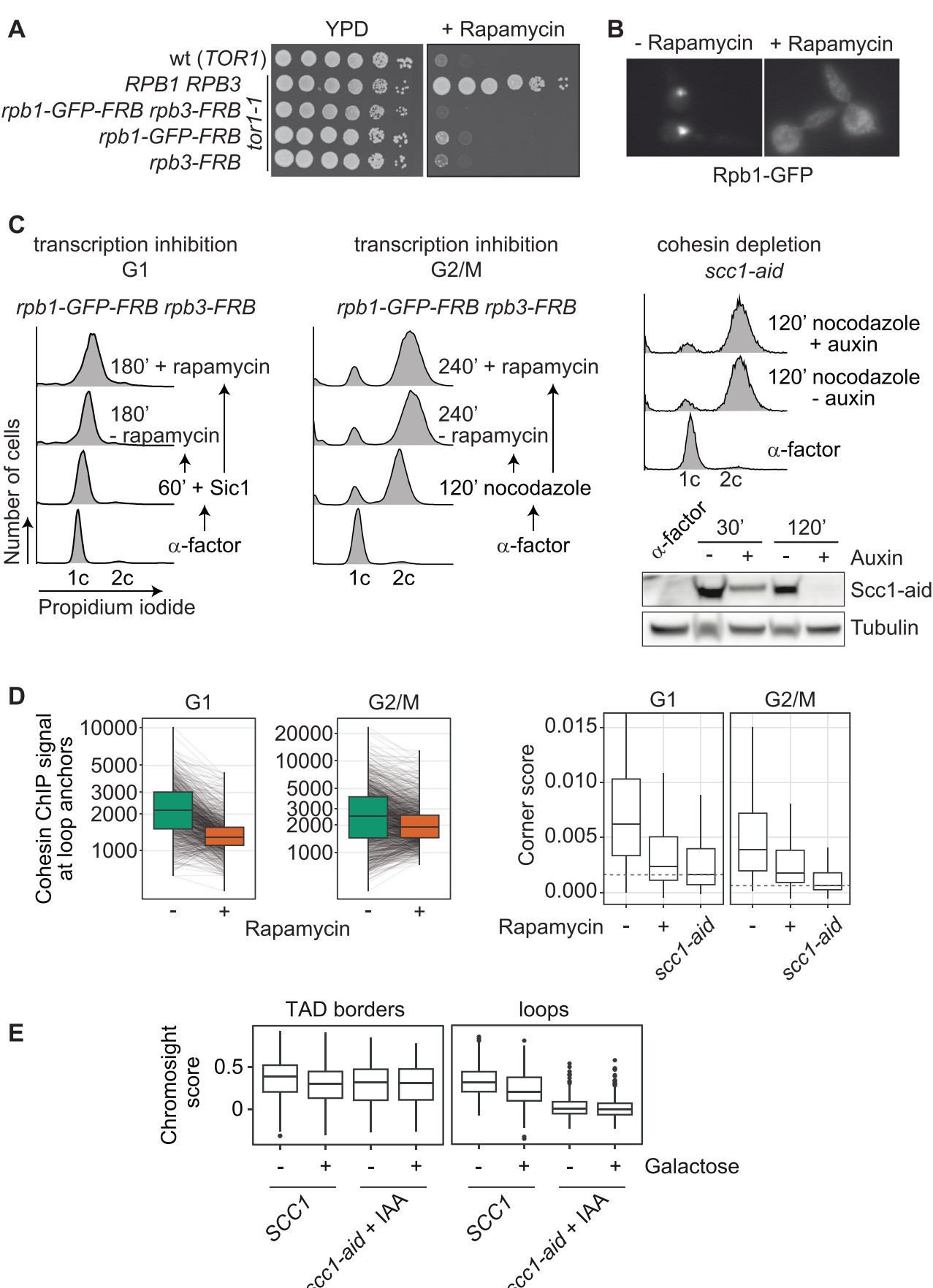

◀ **Figure EV5. Transcription inhibition and its effect on cohesin-mediated chromatin loops.**

(A) 10-fold serial dilutions of cultures of the indicated genotypes were plated onto YPD agar plates, with or without 2 µg/ml added rapamycin, and grown at 30 °C for 2 days. A strain in which both Rpb1 and Rpb3 subunits of RNA polymerase II were fused to FRB showed a tighter response to rapamycin, as compared to strains with either one of the fusions. (B) An example of Rpb1-GFP-FRB relocation from the nucleus to the cytoplasm after one hour 2 µg/ml rapamycin treatment. Cells show the typical elongated bud shape of Sic1-induced late G1 arrest (Lopez-Serra et al, 2013). (C) FACS analysis of DNA content of the cells in the experiment shown in Fig. 3, as well as an experimental outline. Western blot analysis confirmed Scc1-aid depletion by its auxin-inducible degron, at 30 min and 120 min (the time of cell harvest) after release from α-factor synchronisation. Scc1 was detected with the α-aid antibody, tubulin served as the loading control and was detected with a mouse monoclonal α-Tub1 antibody (clone TAT-1). (D) Cohesin ChIP signal intensity distributions at loop anchors (normalised mean reads), in the absence or presence of rapamycin, in both the G1 and G2/M synchronised cultures. Grey lines connect individual ChIP signal intensities under the two conditions (G1: $n = 1059$, G2: $n = 1447$). Corner score distributions of the corresponding loops (G1: $n = 788$, G2: $n = 1060$), before and after transcription inhibition, as well as of the same loop positions sampled following Scc1 depletion, are shown alongside. Box plots represent the median (centre), quartiles (box) and range (whiskers). Baseline corner scores in the absence of cohesin are indicated by dashed lines. (E) Chromosight TAD boundary ($n = 822$) and Chromosight loop score ($n = 1300$) distributions, detected in *SCC1* cells grown in glucose (Fig. 5) and recorded from the maps under the indicated experimental conditions. Box plots represent the median (centre), quartiles (box) and range (whiskers).

