## [Peer Review File · The EMBO Journal]

An extrinsic motor directs chromatin loop formation by cohesin

Thomas Guérin, Christopher Barrington, Georgii Pobegalov, Maxim Molodtsov, and Frank Uhlmann

Corresponding author(s): Frank Uhlmann (frank.uhlmann@crick.ac.uk)

Review Timeline:

Submission Date:	5th Mar 24
Editorial Decision:	12th Mar 24
Revision Received:	2nd Jun 24
Editorial Decision:	25th Jul 24
Revision Received:	30th Jul 24
Accepted:	31st Jul 24

Editor: Hartmut Vodermaier

Transaction Report: Please note that the manuscript was transferred from another journal where it was originally reviewed. Since the original reviews are not subject to EMBO's transparent review process policy, they cannot be published.

Dr. Frank Uhlmann
The Francis Crick Institute
Chromosome Segregation Laboratory
1 Midland Road
London NW1 1AT
United Kingdom

12th Mar 2024

Re: EMBOJ-2024-117178
Cohesin chromatin loop formation by an extrinsic motor

Dear Frank,

Thank you again for submitting your previously reviewed manuscript on loop extrusion-deficient yeast cohesin to The EMBO Journal. I have now had the chance to assess your preliminary responses to the original reports, and concluded that we would be happy to consider an adequately revised version further for publication.

I am therefore inviting you to prepare a new version according to the provided revision plan, and to resubmit it via the link below. When finalizing the study for resubmission, please try to adhere to the guidelines listed below and in our Guide to Authors as closely as possible, as this should greatly facilitate our assessment at the time of resubmission - in particular regarding the completion of our author checklist, the inclusion of editable text files and individual figure files, inclusion of Material and Methods in the main text, conversion of "supplemental" material into Expanded View and/or Appendix content, and reference formatting according to EMBO Journal style (alphabetical order, not numbered). As discussed, please also capitalize on the more extended format of EMBO Journal articles, especially the possibility of including more than 4 main figures.

Finally, please do not hesitate to contact me with any issues that should come up during the revision work, or in case you should require more time than our standard three months - our 'scooping protection' (meaning that competing work appearing elsewhere in the meantime will not affect our considerations of your study) would of course remain valid also during an extended revision.

Thank you again for the opportunity to consider this study for The EMBO Journal. I look forward to receiving your revision.

With kind regards,

Hartmut

3) Revised manuscript text (including main tables, and figure legends for main and EV figures) has to be submitted as editable

text file (e.g., .docx format). We encourage highlighting of changes (e.g., via text color) for the referees' reference.

4) Each main and each Expanded View (EV) figure should be uploaded as individual production-quality files (preferably in .eps, .tif, .jpg formats). For suggestions on figure preparation/layout, please refer to our Figure Preparation Guidelines:

8) Please note that supplementary information at EMBO Press has been superseded by the 'Expanded View' for inclusion of additional figures, tables, movies or datasets; with up to five EV Figures being typeset and directly accessible in the HTML version of the article. For details and guidance, please refer to:

embopress.org/page/journal/14602075/authorguide#expandedview

9) Digital image enhancement is acceptable practice, as long as it accurately represents the original data and conforms to community standards. If a figure has been subjected to significant electronic manipulation, this must be clearly noted in the figure legend and/or the 'Materials and Methods' section. The editors reserve the right to request original versions of figures and the original images that were used to assemble the figure. Finally, we generally encourage uploading of numerical as well as gel/blot image source data; for details see: embopress.org/page/journal/14602075/authorguide#sourcedata

At EMBO Press, we ask authors to provide source data for the main manuscript figures. Our source data coordinator will contact you to discuss which figure panels we would need source data for and will also provide you with helpful tips on how to upload and organize the files.

Further information is available in our Guide For Authors:

In the interest of ensuring the conceptual advance provided by the work, we recommend submitting a revision within 3 months (10th Jun 2024). Please discuss the revision progress ahead of this time with the editor if you require more time to complete the revisions. Use the link below to submit your revision:

Link Not Available

We thank A. Papantonis and the two anonymous reviewers for their constructive critique of our manuscript. Please find their comments copied below, as well as our response how we have addressed these comments in our revised manuscript. We would in advance like to thank the *EMBO Journal* for taking note of these reviews, and for their time appraising the merit of our manuscript and of our reviewer responses.

Reviewer: 1

The manuscript by Guerin and colleagues makes a (perhaps) daring hypothesis for how chromatin loops form via the action of yeast cohesin complexes, and the use simple yet elegant experiments to test it -- and eventually prove it. I enjoyed reading the manuscript, it is clear, well written, experiments are presented in a nuanced way and well controlled. I honestly believe that this is an important finding that merits immediate publication, and I only have two comments/suggestions to make that I hope the authors will consider; both concern the Discussion section.

First, it would be useful if the authors added a small paragraph to discuss their findings in relation to cohesin loop extrusion in mammalian genomes. The more compact yeast genome (and the fruit fly one, for example) have been shown to be driven strongly by transcription as regards 3D chromatin folding. This remains more debated for the human/mouse genomes despite recent evidence (Zhang, Uebelmesser et al, Nat Genet 2023; Barshad et al, Nat Genet 2023). Readers would then benefit from a theoretical extrapolation of the yeast findings to mammalian 3D genome folding, which is somewhat blurred in the current version of the Discussion.

We have taken the opportunity and expanded our discussion on how our observations in the yeast model relates to mammalian genome architecture. Mammalian cohesin loading onto chromatin, like budding yeast cohesin, is promoted by transcription. Following loading, also mammalian cohesin is moved along the genome by transcription. Apart from these similarities, there are interesting distinctions. E. g. mammalian cohesin pauses at polar CTCF binding sites during translocation. Another interesting difference, described in the recent Zhang, Uebelmesser et al publication from the Papantonis laboratory, is an increased sensitivity to transcription inhibition of enhancer-promoter contacts. This publication is now cited in our revised manuscript.

Second, and along similar lines, the statement "studies that reported a transcription requirement for loop formation used increased inhibitor concentrations or combined more than one tool to suppress transcription (39, 40)" is not fully accurate. On the one hand, ref. 39 mostly uses Hi-C and a single depletion approach (i.e., auxin-mediated degradation of RPB1); on the other hand, there is a newer study where Micro-C is used (and therefore closer to the findings of this study, methodologically) with the same 'degron' cell line and in which near-complete RNAPII depletion affects a subset of loops -- those involving active enhancers and promoters [the Zhang, Uebelmesser, et al, Nat Genet 2023 study]. I actually find that there are no discrepancies between these studies, and would like to reiterate that the authors' view on this would be useful to the broad readership this paper will attract.
A. Papantonis

We acknowledge that the Zhang, Uebelmesser, et al study used single auxin-mediated degradation of RPB1 to reveal loss of a subset of cohesin-mediated chromatin folding interactions. We have revised our discussion to include a discussion and citation of this study.

Reviewer: 2

The manuscript by Guerin et al reports a series of biochemical and genetic experiments to test whether the loop extrusion (LE) activity of cohesin described in vitro is indeed responsible for the formation of DNA loops and topologically associating domains (TADs) in vivo. The authors first show that cohesin mutants defective in their LE activity but not in DNA capture in vitro can support cell viability and cohesin-linked loop formation in vivo. Next, they provide evidence that transcription promotes cohesin-linked loop formation and that transcription enlarges pre-existing loops. Finally, they show that TADs can be formed even in the absence of cohesin. Taken together, the authors argue against the widely accepted view that the LE activity of cohesin drives the formation and expansion of chromatin loops in vivo.

Based largely on single-molecule experiments (in vitro) and Hi-C-derived data (in vivo), the model that the LE activity of cohesin drives the formation and expansion of chromatin loops and TADs in vivo has become very popular in the field. This reviewer, however, fully agrees with the authors' argument that in vivo data supporting this model remain scarce. Although the currently available Hi-C data are not inconsistent with the model, it is fair to say that they are often overinterpreted in the field. The current study aims to critically address this important issue and to provide evidence arguing against the above model. If the presented approaches and the derived data are fully convincing, the current manuscript would be a good candidate for publication in Science. However, several questions and concerns need to be clarified before it is considered for publication, as described below:

Major comments:

(1) Is budding yeast the best model system to address this specific question? During the past several years, it has become clear that there are two functionally distinct populations of cohesin, namely, looping cohesin (involved in interphase chromosome organization from G1 to G2) and cohesive cohesin (involved in sister chromatid cohesion from S to G2/M). In animal cells, it is possible to distinguish between the two populations by using various tricks (e.g., sororin depletion). To this reviewer's knowledge, however, the distinction between the two populations of cohesin is much less clear in budding yeast: the chromosomal level of cohesin, if any, is very low in G1 and there is no clear distinction of chromosome binding sites between the two populations. This problem makes the interpretation of Figure 1C and Figure 2 rather difficult. The authors should clarify this important point.

Budding yeast, in our view, is a suitable model to investigate the nature of cohesin-mediated chromatin loops. This model organism displays a pronounced set of cohesin-dependent chromatin loops that are amenable to experimental scrutiny. The reviewer's perceived drawback of the model stems from a misunderstanding around budding yeast G1. 'G1' is

often used to describe a pheromone induced 'G0' arrest when cohesin is indeed largely absent from chromosomes. To overcome this limitation, we use a genetically engineered 'late G1' arrest in which the SBF-controlled G1/S transcription programme is active, cohesin is expressed, and loaded onto chromosomes. Our G1 cells therefore allow us to study 'looping cohesin', while our G2/M cells, like mammalian cells, contain a mixture of looping and cohesive cohesin. The 'late G1' state used in our experiments is better introduced in the revised manuscript.

(2) As an impressive prelude to the whole story, the authors first describe loop extrusion deficient (but capture-competent) mutants of cohesin in Figure 1. Curiously, however, they are only used in Figure 1 but not in subsequent figures. This reviewer wonders if the mutant cohesins could be used more effectively in other experiments, too, to further their arguments.

We agree with the reviewer that our loop extrusion deficient mutants from Figure 1 could be used in subsequent figures. Indeed, Expanded View Figure 2 contains a broad range of experiments to characterise cells harbouring loop extrusion deficient cohesin. During our revisions, we have added the analysis of meiotic cell divisions with loop extrusion deficient cohesin.

(3) The extrusion-deficient mutations described here are not complete separation-of-function mutations: they are hypomorphic in terms of DNA binding and DNA capture in vitro (Fig S1). These properties make the interpretation of other experiments not straightforward (e.g., Fig 1C and Fig S2D).

The reviewer is right in that Smc1-4E cohesin is a hypomorph with regards to loop extrusion. However, instead of complicating the interpretation, together with the stronger loss of function Scc3-3E cohesin, such an 'allelic series' of mutations of increasing severity is a powerful genetic tool. Smc1-4E cohesin shows residual *in vitro* loop extrusion activity, but shows weaker *in vivo* loop formation than Scc3-3E cohesin. Conversely, DNA capture is more strongly affected in Smc1-4E cohesin than Scc3-3E cohesin. Thus, *in vivo* chromatin loop formation quantitatively correlates with cohesin's DNA capture capability, more so than with its loop extrusion activity.

(4) This reviewer would like to know to what extent the LE activity of wt and mutant cohesins is sensitive to experimental conditions (e.g., different salt concentrations).

As part of our revision experiments, we have studied loop extrusion at different salt concentrations. We could observe *in vitro* loop extrusion also under physiological ionic strength conditions, and the effects of our loop extrusion mutations was similar throughout all these experimental conditions. Therefore, if loop extrusion occurs *in vivo*, we can be confident that our mutations should have affected the process.

(5) The analysis presented in Fig 4C requires quantification of loops and TADs.

Quantification of loops and TAD boundary strengths, in what is now Figure 5D, have been added during the revisions.

Typo:

Page 5, line 20: Fig. 3D > Fig. 2D?

Page 7, line 3, Fig. 4B > Fig. S4B?

We thank the reviewer for alerting us to these typos, which have been corrected.

Reviewer: 3

The manuscript Guerin and colleagues studies how cohesin contributes to DNA loops in budding yeast. The authors generate cohesin mutants that have lost, or partially lost, the ability to extrude loops in vitro. When introduced in vivo, these mutants do not abolish loops, but chromatin contacts persist, or are only partially reduced. The authors build on these observations and argue that cohesin forms loops via a mechanism other than loop extrusion, and that transcription is the extrinsic motor that promotes cohesin-dependent loops.

This manuscript includes interesting findings, but in my view also includes too many bold statements that I simply cannot reconcile with the presented data. The interpretation of the data to me is too much in favor of only one model, while neglecting, or not addressing, relevant literature and earlier work. In my view, this makes the manuscript unsuited for publication in Science.

Below I provide some suggestions that I hope will be helpful for submission elsewhere:

1) The title “Cohesin loop formation by an extrinsic motor” is tantalizing, but does not do justice to the findings presented in the paper. “Transcription promotes cohesin loop expansion in yeast” would be a better title, as this is closer to what the data actually shows.

Our experiments suggest that transcription, as an extrinsic motor, both enables the formation of chromatin loops by cohesin, as well as promotes their expansion. As part of our revisions, we therefore suggest the revised title to read: “An extrinsic motor directs cohesin chromatin loop formation and expansion”.

2) It is important to consider that the in vitro loop extrusion assays may not be representative of what happens in cells (Fig 1B and C). The mutated residues in the cohesin subunits have been described to play a role in loop extrusion in a manner that is independent of ATPase activity. These residues have been proposed to affect translocation on DNA. These mutants likely bind the DNA and in the in vitro setting may be unable to walk on the DNA strand by themselves. In vivo however, it has been proposed that certain factors can anchor cohesin on chromatin and stimulate loop enlargement by e.g. allowing cohesin to translocate. The in vitro system doesn't contain these elements that may be 'supporting' cohesin to build loops, still via loop extrusion.

The reviewer makes a pertinent point that cohesin, mutant or not, might receive help when walking along the DNA strand *in vivo*. It is the purpose of our study to investigate where such help might be coming from, and our results suggest that transcription provides this help.

3) Related to point 2, it would be important to repeat the loop extrusion experiments with different buffer conditions. Introducing mutations that change cohesin's electrostatic interactions with the DNA may require optimization of the buffer conditions, such as different salt concentration etc.

As part of our revision experiments, we have studied loop extrusion at different salt concentrations. We could observe *in vitro* loop extrusion also under physiological ionic strength conditions, and the effects of our loop extrusion mutations was similar throughout all these experimental conditions. Therefore, if loop extrusion occurs *in vivo*, we can be confident that our mutations should have affected the process.

*4) The Smc1 4E mutant has reduced ability to capture DNA by half (Fig S1.D) and was described by Bauer et al., 2021 to have its ATPase activity reduced to $\pm 40\%$. It is expected that this mutant shows less effective *in vitro* loop extrusion and as a consequence displays less pronounced loops in cells. The Scc3E mutant in this case is much more informative.*

The reviewer is right in that Smc1-4E cohesin is a hypomorph with regards to loop extrusion. However, instead of complicating the interpretation, together with the stronger loss of function Scc3-3E cohesin, such an 'allelic series' of mutations of increasing severity is a powerful genetic tool. Smc1-4E cohesin shows residual *in vitro* loop extrusion activity, but shows weaker *in vivo* loop formation than Scc3-3E cohesin. Conversely, DNA capture is more strongly affected in Smc1-4E cohesin than Scc3-3E cohesin. Thus, *in vivo* chromatin loop formation quantitatively correlates with cohesin's DNA capture capability, more so than with its loop extrusion activity.

5) In the micro-C experiments (Fig1.C) it is important to deplete cohesin in the mutants to show that the persistent loops are indeed cohesin-dependent.

We perform cohesin depletion as part of Figures 3 and 5, revealing that chromatin loops observed between cohesin binding sites are indeed cohesin-dependent. This result confirms previous reports by Costantino et al. 2020 and Dauban et al. 2020.

6) Fig2A shows that cohesin depletion in G1 arrested cells did not result in TAD loss. This statement would be stronger if supplemented by aggregate TAD analyses on yeast with vs without Scc1.

The persistence of TADs following cohesin depletion is indeed striking. Following from the reviewer's suggestion, as part of our revisions, we have performed aggregate TAD analyses in the presence and absence of Scc1. The results from this analysis are reported in a new Figure panel 5D.

7) In Fig3A the ChIP-microarray traces show that rat1-1 cells at 37C lose certain contacts, but not all. It is interesting to note that the persistent contacts colocalize with longer convergent genes, whereas the contacts at shorter convergent genes are lost. Could the authors speculate on the reason why? It would be beneficial to perform ChIP for RNA

Pol II to assess whether polymerases indeed colocalize with the Scc1 peaks, and assess whether RNA transcripts are indeed from these loci to test whether these genes are continuously transcribed in rat1-1 cells at 37C.

Changes to the transcriptional programme following Rat1 inactivation have been previously documented by Baejen et al. 2017. As part of our revisions, we have used the resources provided by these authors to investigate how transcriptional changes following Rat1 inactivation correlate with cohesin looping pattern changes. These analyses revealed that cohesin binding sites that will be displaced by Rat1 inactivation showed a striking exclusion of RNA polymerase II before inactivation, but were covered by the transcription machinery after Rat inactivation. These findings are reported in a new Figure panel EV4A.

8) In Fig3B it would be helpful to include quantification of loop scores, like in 1C or 2B.

In the revised figure (now Figure 2B), we have included mean corner scores in the loop aggregate plots.

9) If one compares the micro-C from Fig3C in rat1-1 cells at 25C with the matrix in Fig3D at G2, 25C - technically the only difference between these conditions is Scc2 depletion. In these example matrices, it looks as if the loops disappeared upon Scc2 loss, which is inconsistent with the presented model. Perhaps quantification of these contacts would be helpful to avoid confusion?

It is correct that cohesin-dependent chromatin loops become weaker following Scc2 depletion, most likely because dynamic loops are now lost. A lower level of stable loops remains detectable. These observations are consistent with what was observed by Bastié et al. 2022. Our experiment studies the behaviour of these remaining, yet clearly detectable, loops in response to Rat1 inactivation.

10) The data showing that TADs are formed in the absence of cohesin (Fig 4A), and as result of transcription, would be stronger if more examples were included. The example of one locus is not very powerful.

Galactose addition induces transcription of a small set of genes, amongst which the bidirectional *GAL7-GAL10-GAL1* locus shows the most pronounced response. A recently posted preprint (<https://doi.org/10.1101/2023.12.29.573667>) additionally reports cohesin-independent TAD formation following galactose addition at the *GAL2* locus, and we have referred to this study in our revised manuscript. As for our own analysis of the *GAL7-GAL10-GAL1* locus, we have added a domain boundary analysis as a new Figure panel 5C, which clearly and quantitatively documents insulation that arises as the consequence of transcriptional induction.

11) Information on the number of reads, quality controls, comparison of replicates, etc. of the micro-C analyses should be included in the methods/supplements.

These items of information have been added as Appendix Table S2 and Appendix Figure S1, respectively, to the revised manuscript.

12) *The proposed model in Fig4C is barely explained in the manuscript. Please elaborate. For example, it is not clear why Scc2/Scc4 is included in the schematic, since the authors argue that cohesin does not intrinsically power its own movement on DNA but rather is supported by RNA polymerase in the absence of Scc2.*

In our revised manuscript, we provide a better explanation of our model, now Figure 6, including an informative Figure legend. We have clarified how Scc2 supports initial loop capture but, unlike in the loop extrusion model, is dispensable for subsequent loop expansion.

13) *There are several further textual changes to implement in the narrative:*

- *Page 3: "Whether in vivo chromatin loops and TADs indeed form a loop by extrusion has not yet been experimentally tested". It is somewhat weird to phrase it this way, as there is quite a body of work that has from various angles addressed these types of questions. I agree that it is technically challenging to directly show loop extrusion in vivo, but following this logic, the current manuscript doesn't test this either.*

- *Page 4: "We conclude that cohesin complexes that cannot extrude DNA loops (...) nonetheless support yeast growth...". This is too bold a statement. One should say: "... cannot extrude loops in vitro".*

- *Discussion: "Instead of loop extrusion, we find that transcription promotes loop formation...". It has not been shown that it is about the formation of loops. It could also be that small loops are already formed and that transcription then causes their enlargement. So perhaps change the text to 'enlargement'?*

- *Another comment on the above sentence is that transcription promoting loop enlargement and loop extrusion do not need to be mutually exclusive models. The authors should be careful with using words such as "instead" and speculate more openly on interpretation of their data, also taking into account earlier work.*

We have considered all these additional suggestions while preparing our revised manuscript text and discussion.

Dr. Frank Uhlmann
The Francis Crick Institute
Chromosome Segregation Laboratory
1 Midland Road
London NW1 1AT
United Kingdom

25th Jul 2024

Re: EMBOJ-2024-117178R
An extrinsic motor directs cohesin chromatin loop formation and expansion

Dear Frank,

Thank you again for submitting your study, revised in response to the previous reviewer comments from another journal. As discussed, we sent it to a trusted arbitrating referee of our own journal, who has now provided the comments copied below. Since our arbitrator is overall supportive of the revised study, we shall be happy to publish it in The EMBO Journal, but would first still invite you to consider/incorporate/respond to the points raised by our referee in the final version and/or the final point-by-point response.

In addition, there are also a few editorial issues that should still be addressed at this point:

- We noted that a control panel in Figure 1C also appears in Figure 5D. Please clarify if it really stems from the same experiment, and if so, please make sure to explicitly state this in the respective figure legends.

- Please adjust the order of the manuscript sections: Title page with complete author information, Abstract, Keywords, Introduction, Results, Discussion, Materials & Methods, Data Availability Section, Acknowledgements, Disclosure and Competing Interests Statement, References, Main figure legends, Tables, Expanded Figure Legends.

- Since our Expanded View Figures are limited to five, please consider turning two of the current seven EV Figures into (parts of) main figures, or into additional Appendix Figures.

- As we are switching from a free-text author contribution statement towards a more formal statement based on Contributor Role Taxonomy (CRediT) terms, please remove the present Author Contribution section and instead specify each author's contribution(s) directly in the Author Information page of our submission system during upload of the final manuscript. See <https://casrai.org/credit/> for more information.

- Please adjust the format for citation of preprints as specified in our author guidelines:

The citation in the text should be: "(preprint: NAME1 et al, YEAR)"

The citation in the reference list: "Author NAME1, Author NAME2, ... (YEAR) article title. bioRxiv doi: XXX"

- Finally, please provide suggestions for a short 'blurb' text prefacing and summing up the study in two sentences (max. 250 characters), followed by 3-5 one-sentence 'bullet points' with brief factual statements of key results of the paper; they will form the basis of an editor-written 'Synopsis' accompanying the online version of the article. You may also upload a dedicated synopsis image (in PNG or JPG format, with the modest dimensions of EXACTLY 550 pixels wide and 300-600 pixels high), but we could in my view also simply re-use the current Figure 6 for this purpose.

I am therefore returning the manuscript to you for a final round of revision, to allow you to make these modifications and upload the revised files. Once we will have received them, we should hopefully be ready to swiftly proceed with formal acceptance and production of the manuscript.

With kind regards,

Hartmut

9) To facilitate reproducibility and cross-laboratory adoption of methodologies, please structure the Materials & Methods section as outlined in our guide to authors, including a completed Reagents and Tools Table that can be downloaded from our author guidelines as well (<https://www.embopress.org/page/journal/14602075/authorguide#structuredmethods>).

10) Digital image enhancement is acceptable practice, as long as it accurately represents the original data and conforms to community standards. If a figure has been subjected to significant electronic manipulation, this must be clearly noted in the figure legend and/or the 'Materials and Methods' section. The editors reserve the right to request original versions of figures and the original images that were used to assemble the figure. Finally, we generally encourage uploading of numerical as well as gel/blot image source data; for details see: embopress.org/page/journal/14602075/authorguide#sourcedata

At EMBO Press, we ask authors to provide source data for the main manuscript figures. Our source data coordinator will contact you to discuss which figure panels we would need source data for and will also provide you with helpful tips on how to upload and organize the files.

Further information is available in our Guide For Authors:

In the interest of ensuring the conceptual advance provided by the work, we recommend submitting a revision within 3 months (23rd Oct 2024). Please discuss the revision progress ahead of this time with the editor if you require more time to complete the revisions. Use the link below to submit your revision:

Link Not Available

Referee #1:

Chromosome conformation capture experiments (HiC) have shown that the genomes from bacterial to man can be folded into loops. In higher species, loops can form topologically associating domains (TADs), especially at cell population levels. There is strong evidence to indicate that chromosome loop formation requires the SMC family of protein complexes. How SMC proteins mediate loop formation in living cells is unclear. A prevailing model is that they do so through a process termed loop extrusion. Evidence supporting loop extrusion is strong. First, SMC complexes, including condensin, cohesin, and the SMC5/6 complex, are capable of extruding DNA loops in vitro. Second, the CTCF convergence rule (where only head-to-head orientations of CTCF sites can act as loop barriers in vertebrates) is consistent with the loop extrusion model. Obviously, it would be difficult to obtain definitive evidence for loop extrusion in living cells.

In the current study, Uhlmann and coworkers designed two budding yeast cohesin mutants that were defective in DNA binding and in extruding DNA loops in vitro. They showed that these mutants still supported loop formation in budding yeast cells based on HiC. They also showed that perturbing transcriptional termination can enlarge cohesin-dependent DNA loops in cells and that strong transcriptional activity suffices to produce TAD boundaries. Overall, these results indicate that DNA looping by cohesin in vivo involves mechanisms that are different from those observed in vitro. It is also apparent that transcription can influence cohesin-dependent chromosome looping. As such, the results are exciting and should be reported in a major journal.

On the other hand, I agree with reviewer 3 that the conclusions are overstated. Even with the modifications to the text, I still feel that conclusions need to be toned down further. My specific comments are listed below for the authors to consider.

Major points

(1) The claim that an extrinsic motor propels cohesin in vivo is not supported by strong data. The most important activity of cohesin is its ATPase. If the author can show that the ATPase-dead mutant of cohesin can still be pushed along chromosomes, this would lend strong support to their claim. Of course, ATPase-deficient cohesin mutants cannot be loaded onto DNA, making it impossible to test this possibility experimentally. DNA-binding deficient mutants can in theory be compensated by other chromatin-binding or cohesin-binding factors in yeast cells. This major caveat, as noted by the authors, cannot be completely ruled out. Therefore, I would encourage the authors to further tone down their conclusions.

(2) There is strong evidence in the literature to suggest the existence of two types of cohesin in cells: NIPBL/SCC2-bound cohesin capable of loop extrusion and the PDS5-bound cohesin that trap DNA topologically. It is possible that loop expansion caused by transcriptional termination defects is driven by PDS5-bound cohesin. If so, the loop expansion should be PDS5 dependent. Ideally, the authors should test this possibility experimentally. Minimally, they should discuss this possibility.

(3) The section on transcriptional intermediates as substrates for cohesin capture is highly speculative. Topoisomerase inactivation is expected to produce pleiotropic effects, which are not limited to transcription. This experiment is not a strong support for unwound DNA at transcriptional sites as substrates for cohesin capture. More evidence is needed to substantiate this claim.

Minor points

(1) Why does SMC1-4E cohesin extrude DNA loops with faster rate, as compared to WT cohesin? This point should be briefly discussed.

(2) The TAD boundaries are not clear in Fig. 5A and 5B. Can the authors mark the boundaries? Why does the upper right corner of Fig. 5A appear different from the rest of the map?

Referee #1 (Report for Author)

Chromosome conformation capture experiments (HiC) have shown that the genomes from bacterial to man can be folded into loops. In higher species, loops can form topologically associating domains (TADs), especially at cell population levels. There is strong evidence to indicate that chromosome loop formation requires the SMC family of protein complexes. How SMC proteins mediate loop formation in living cells is unclear. A prevailing model is that they do so through a process termed loop extrusion. Evidence supporting loop extrusion is strong. First, SMC complexes, including condensin, cohesin, and the SMC5/6 complex, are capable of extruding DNA loops in vitro. Second, the CTCF convergence rule (where only head-to-head orientations of CTCF sites can act as loop barriers in vertebrates) is consistent with the loop extrusion model. Obviously, it would be difficult to obtain definitive evidence for loop extrusion in living cells.

In the current study, Uhlmann and coworkers designed two budding yeast cohesin mutants that were defective in DNA binding and in extruding DNA loops in vitro. They showed that these mutants still supported loop formation in budding yeast cells based on HiC. They also showed that perturbing transcriptional termination can enlarge cohesin-dependent DNA loops in cells and that strong transcriptional activity suffices to produce TAD boundaries. Overall, these results indicate that DNA looping by cohesin in vivo involves mechanisms that are different from those observed in vitro. It is also apparent that transcription can influence cohesin-dependent chromosome looping. As such, the results are exciting and should be reported in a major journal.

On the other hand, I agree with reviewer 3 that the conclusions are overstated. Even with the modifications to the text, I still feel that conclusions need to be toned down further. My specific comments are listed below for the authors to consider.

Major points

(1) The claim that an extrinsic motor propels cohesin in vivo is not supported by strong data. The most important activity of cohesin is its ATPase. If the author can show that the ATPase-dead mutant of cohesin can still be pushed along chromosomes, this would lend strong support to their claim. Of course, ATPase-deficient cohesin mutants cannot be loaded onto DNA, making it impossible to test this possibility experimentally. DNA-binding deficient mutants can in theory be compensated by other chromatin-binding or cohesin-binding factors in yeast cells. This major caveat, as noted by the authors, cannot be completely ruled out. Therefore, I would encourage the authors to further tone down their conclusions.

We agree with the reviewer, it would be hugely informative to modulate the cohesin ATPase in a conditional manner to study its contribution to the various stages of cohesin function. As yet, there isn't an equivalent of, say, an ATP analog sensitive SMC allele, though it might be possible to obtain such reagents in the future. It is also correct that Straight et al. 1996 have used a tetramerizing form of the GFP–Lac repressor to provide cohesin-independent sister chromatid cohesion. However, this tool would still fall short of allowing the study of ATPase deficient cohesin, bound to chromosomes. In the absence of such tools, we have followed the reviewer's advice and have toned down our conclusions.

(2) There is strong evidence in the literature to suggest the existence of two types of cohesin in cells: NIPBL/SCC2-bound cohesin capable of loop extrusion and the PDS5-bound cohesin that trap DNA topologically. It is possible that loop expansion caused by transcriptional termination defects is driven by PDS5-bound cohesin. If so, the loop expansion should be PDS5 dependent. Ideally, the authors should test this possibility experimentally. Minimally, they should discuss this possibility.

The idea of two types of cohesin stems from provisional biochemical analyses reported by Murayama et al. 2015 and Petela et al. 2018. Because Pds5-containing cohesin is unable to perform *in vitro* loop extrusion, the idea of ‘Scc2-only’ and ‘Pds5-only’ cohesin has taken hold. However, there is as yet little evidence-based clarity around cohesin complex composition in live cells. E.g. it is unclear whether Scc2-only cohesin indeed exists *in vivo*. We refer these important questions to be resolved in future work.

(3) The section on transcriptional intermediates as substrates for cohesin capture is highly speculative. Topoisomerase inactivation is expected to produce pleiotropic effects, which are not limited to transcription. This experiment is not a strong support for unwound DNA at transcriptional sites as substrates for cohesin capture. More evidence is needed to substantiate this claim.

The reviewer is right, of course, that net positive helical tension introduced by *TopA* will affect multiple chromatin processes, e.g. histone turnover is likely compromised. We note that the *TopA* experiment is used solely as supporting evidence, not as the basis for making a new claim. Nevertheless, in line with the reviewer’s recommendation, we have added a note of caution around the expected pleiotropic effects of *TopA* in this experiment.

Minor points

(1) Why does SMC1-4E cohesin extrude DNA loops with faster rate, as compared to WT cohesin? This point should be briefly discussed.

The reviewer makes an interesting observation. Indeed, Smc1^{4E}-cohesin shows a greater median extrusion rate compared to wild type cohesin. This skew could arise if the small number of loop extrusion events by this variant are biased towards DNAs under low tension, on which extrusion might proceed faster. While loop extrusion by Smc1^{4E}-cohesin remains to be characterised in further detail, we mention this possible explanation in the corresponding Fig. EV1B legend.

(2) The TAD boundaries are not clear in Fig. 5A and 5B. Can the authors mark the boundaries? Why does the upper right corner of Fig. 5A appear different from the rest of the map?

The TAD boundaries coincide with the edges of the *GAL7-GAL10-GAL1* gene cluster, which is indicated in the figure and now better explained in the figure legend. The ‘upper right corner’ appears white due to the insulation provided by the *GAL7-GAL10-GAL1* gene cluster following transcription activation. This effect was recently also reported in a preprint by Chapard et al. 2023, using the complementary Hi-C approach.

Dr. Frank Uhlmann
The Francis Crick Institute
Chromosome Segregation Laboratory
1 Midland Road
London NW1 1AT
United Kingdom

31st Jul 2024

Re: EMBOJ-2024-117178R1
An extrinsic motor directs chromatin loop formation by cohesin

Dear Dr. Uhlmann,

Thank you for submitting your final revised manuscript for our consideration. I am pleased to inform you that we have now accepted it for publication in The EMBO Journal.

Yours sincerely,

Hartmut Vodermaier
